# Change in negative emission burden between an overshoot versus peak-shaved Stratospheric Aerosol Injections pathway

Susanne Baur[1], Benjamin M. Sanderson[2], Roland Séférian[3], Laurent Terray[1]

[1]CECI, Université de Toulouse, CERFACS, CNRS, Toulouse, France

[2]Centre for International Climate and Environmental Research (CICERO), Oslo, Norway

[3]CNRM, Université de Toulouse, Météo-France/CNRS, Toulouse, France

*Correspondence to*: Susanne Baur (susannebaur1796@gmail.com)

**Abstract.** Stratospheric Aerosol Injection geoengineering (SAI) is being investigated as a potential means of temporarily reducing the impact of global warming, allowing additional time for the implementation of climate mitigation strategies. SAI operates by intervening in the radiative energy balance of the Earth system, exerting a temporary direct cooling effect on the climate. However, SAI also indirectly affects global temperature through its impact on atmospheric $CO_2$ levels by influencing the natural carbon uptake efficiency. Most previous research on the carbon cycle under SAI suggests that continuous injections enhance the uptake of carbon, implying a larger number of allowable emissions for a given temperature target relative to a simulation without SAI. However, there are considerable uncertainties regarding the extent and timeline of facilitation or inhibition of atmospheric carbon removal under SAI. In this study, we evaluate the extent of change in negative emission burden over the entire trajectory of a peak-shaving SAI deployment (SSP534-sulfur) compared to the baseline overshoot pathway (SSP534-over) that does not involve SAI. We run the SSP534-over scenario on the CNRM-ESM2-1 Earth System Model from 2015 to 2249 and compare it to the simulation where, under SSP534-over conditions, SAI is used to maintain 2°C warming (SSP534-sulfur). The results indicate that carbon effects are reinforced under SAI. While the land carbon reservoir is a carbon sink, SAI enhances the uptake further; when the land acts as a carbon source, SAI enhances the outgassing. Thereby, carbon fluxes associated with SAI evolve over time: The increase in carbon uptake under SAI during the positive emission phase confirms prior studies and substantiates the concept of buying time during SAI ramp-up, later stages of the peak-shaved SAI scenario show the carbon benefit reducing and turning into an additional obstacle making a phase-out of SAI more difficult by enhancing the carbon removal burden. The findings of this study may be contingent upon the configuration of the injection design and the representation of SAI within the model, as well as the underlying overshoot scenario. Further research is necessary to validate these results using different models incorporating diverse SAI deployment strategies and underlying emission trajectories.

# 1 Introduction

Solar Radiation Modification (SRM) is increasingly being discussed as a potential temporary approach to lower global mean temperature while mitigation efforts, such as greenhouse gas emission reductions and atmospheric Carbon Dioxide Removal (CDR), are being sufficiently scaled up (Climate Overshoot Commission, 2023; NASEM, 2021). A commonly used framework is the so-called "peak-shaving" framework where SRM is used on top of an overshoot-pathway to avoid global warming from surpassing the given threshold (MacMartin et al., 2018; Sugiyama et al., 2018; WMO, 2022). The primary intended cooling

effect from SRM comes from directly modifying the radiative energy imbalance of the Earth system. However, indirectly, SRM changes global surface air temperature through its impact on the airborne fraction of $CO_2$ by influencing the natural carbon uptake efficiency of the two big carbon reservoirs, land and ocean. Most previous research on the carbon cycle and SRM indicates that continuous Stratospheric Aerosol Injection (SAI), one type of SRM, enhances the global uptake of carbon by land and ocean (e.g. Muri et al., 2018; Plazzotta et al., 2019; Tjiputra et al., 2016; Xia et al., 2016). However, there are

considerable uncertainties regarding the extent of the carbon cycle reinforcement and the timeline of the response (Plazzotta et al., 2019) and hence the extent and timeline of facilitation or inhibition of atmospheric carbon removal under SRM.

SAI could affect marine and terrestrial carbon uptake in several ways. On land, carbon uptake is governed by changes in plant photosynthesis in combination with alterations to autotrophic and heterotrophic respiration. Impacts to plant photosynthesis occur under SRM due to conditions of high atmospheric $CO_2$, low ambient temperatures and changes in radiation reaching the

plants' leaves. The impact of high atmospheric $CO_2$ on plants, so-called "$CO_2$ -fertilization"-effect, has been found favourable for photosynthesis in several studies on SRM when comparing to a scenario where mitigation is used to lower temperatures rather than SRM (e.g. Duan et al., 2020; Glienke et al., 2015; Govindasamy et al., 2002; Kravitz et al., 2013; Yang et al., 2020). At the same time, when SRM is compared to a scenario with the same background emissions but no SRM, lower temperatures decrease heat stress on plants which promotes additional carbon uptake (Jin and Cao, 2023; Kravitz et al., 2013;

Tilmes et al., 2020) but are disadvantageous for ecosystems in higher latitudes or mountainous regions where the low temperatures are a limit to plant growth (Glienke et al., 2015; Tilmes et al., 2020; Xia et al., 2016; Zhang et al., 2019). And, low temperatures can reduce soil nitrogen mineralization which in turn inhibits the $CO_2$ fertilizing effect on plant photosynthesis (Duan et al. 2020). In addition to ambient temperature and $CO_2$ concentration, SAI would affect photosynthesis by altering the ratio of direct to diffuse radiation that reaches the plants' surface (Xia et al., 2016). The increased number of

aerosols from SAI enhances the amount of diffuse radiation that reaches the surface while decreasing the amount of direct light. This "diffuse-light fertilization"-effect can enhance productivity in certain types of ecosystems because it allows shaded leaves to absorb more light (Gu et al., 2002; Xia et al., 2016) as evidenced in the Amazon Rainforest from increased diffuse radiation from biomass burning (Rap et al., 2015). However, Kalidindi et al. (2015) and Duan et al. (2020) found that the effect of the total radiation reduction might offset the increase in shaded productivity. Lastly, in Duan et al. (2020), H. Lee et al.

(2020) and Muri et al. (2018) the modified hydrological cycle under SAI significantly affected the photosynthesis of plants. In addition to modifying photosynthesis, SAI affects land carbon storage by altering plant and soil respiration, i.e. the process of

carbon release, as lower ambient temperatures reduce heterotrophic and autotrophic respiration (Jin and Cao, 2023), especially when compared to the same emission baseline without SRM-induced temperature reductions. The difference in hydrological processes can change soil moisture content which also affects soil respiration (Yan et al., 2018). Furthermore, lower regional

temperatures under SRM compared to the same emission baseline without SRM and a reduction in wind speeds over most land regions compared to both an unmitigated emission baseline or a mitigated world with the same global temperature mean as with SRM (Baur et al., 2024a; Tang et al., 2023) may cause less disturbance to the land carbon reservoir through forest fires, wind throw or floods.

Several studies find a less pronounced impact of SAI on the ocean carbon uptake in comparison to its impact on the terrestrial

sphere (Jin et al., 2022; Jin and Cao, 2023), other studies have identified the opposite effect (Muri et al., 2018; Tjiputra et al., 2016). In the ocean the principal drivers are the increased $CO_2$ solubility into seawater and the impacts on the ocean biological pump compared to a scenario with the same emission baseline but no SRM (Tjiputra et al., 2016). $CO_2$ solubility into seawater is enhanced due to the lower sea surface temperatures and modified ocean hydrodynamics, such as stratification and currents, with SAI. The biological pump is sensitive to sea surface temperatures (Kwiatkowski et al., 2020) and light availability but

the net effect of marine ecosystems to a change in climate is influenced by local physical and biogeochemical conditions (Lauvset et al., 2017), which can vary between different regions and ocean model settings. In the Arctic, for example, SAI reduces oceanic $CO_2$ uptake because the larger sea ice cover under SRM than in a scenario with the same emission profile but no SRM inhibits $CO_2$ uptake (Jin and Cao, 2023; Tjiputra et al., 2016). In a multi-model study, Plazzotta et al. (2019) and Muri et al. (2018) find an increase in total carbon uptake under SAI compared to both the high-warming emission baseline but no-

SRM scenario and the scenario that uses mitigation to avoid warming instead of SRM, with reduced sea surface temperatures being the main driver of the response when compared to the high-warming scenario and higher atmospheric $CO_2$ concentration the main driver when comparing to the mitigated world. Jin and Cao (2023) report a slight reduction in global oceanic carbon uptake under SAI compared to the emission baseline no SRM scenario which they attribute to the combined influence of lower sea surface temperatures, which enhance $CO_2$ uptake, and lower atmospheric $CO_2$ from enhanced land carbon uptake, which

reduce marine $CO_2$ uptake (Jin et al., 2022; Jin and Cao, 2023). Regarding marine biogeochemical changes under SAI, Lauvset et al. (2017) found reductions in the biological pump due to reduced shortwave radiation reaching the oceans' surface layers which lowers phytoplankton growth rates. Using the same model but a different SAI setup, these results have been confirmed by Tjiputra et al. (2016). While $CO_2$ solubility into seawater and the biological pump represent the primary drivers of the response, the simulated ocean carbon uptake is additionally sensitive to the evolution of $CO_2$ in the scenario, the modeling

setup of prescribed or prognostic atmospheric $CO_2$ and the baseline oceanic stratification, often resulting in little consensus between models.

Most of the aforementioned results are based on climate projections that extend until the end of this century. With that they only cover the time of SRM deployment from initialization to high deployment, and occasionally include a sudden termination

of SRM. The timing and total magnitude of carbon uptake by the reservoirs over the entire period of an overshoot, i.e. a peak-shaving setup, is yet unclear. So far, only one study has looked at carbon cycle processes under a peak-shaving framework (Tilmes et al., 2020). However, their simulation also ends at the end of this century, not allowing for a comprehensive analysis of all phases of a peak-shaving framework. Here, the climate and carbon cycle dynamics of the whole overshoot period are explored under an extended overshoot trajectory that goes until 2249. We look at the entire period of a hypothetical SRM peak-shaving deployment in a large climate overshoot scenario: from initialization to max deployment, followed by a phase-out period and 100 years after SRM cessation. The goal of this study is to provide insight into how the modified uptake of atmospheric $CO_2$ by land and ocean under an SAI peak-shaved pathway compared to an overshoot pathway without SRM could change the amount of negative emissions that are required to follow a given atmospheric $CO_2$ trajectory. This question is highly relevant as sink enhancement could lead to a lower peak in atmospheric $CO_2$ concentration, which could be important for atmospheric $CO_2$ sensitive impacts such as ocean acidification, and shorter peak-shaving timescales; sink degradation would prolong the SRM deployment or require higher amounts of CDR and increase the difficulty of phasing SRM out. A reduced negative emission burden (NEB), especially during the initial decades of SAI, could support the framework of using SRM as a tool to buy time for conventional mitigation measures to take effect.

## 2 Methods

### 2.1 Model and simulations

The data underlying this study is the overshoot scenario SSP534-over and its modified version for this study, SSP534-sulfur. SSP534-over is part of the coordinated Coupled Model Intercomparison Project Phase 6 (CMIP6) group of experiments (Eyring et al., 2016; O'Neill et al., 2016) and in this study is used as a baseline on top of which SAI is applied to avoid the temperature overshoot and instead stay at a global mean temperature increase above pre-industrial (1860-1900) of 2°C (Fig 1). This SAI-modified SSP534-over pathway is referred to as SSP534-sulfur in this study. SSP534-over follows the storyline of the Shared Socio-Economic Pathway 5 (SSP5) which is characterized by strong fossil fuel driven economic growth (O'Neill et al., 2016). The scenario assumes no climate policy until the mid 21st century, followed by late and intense mitigation action, with an emissions peak and emission cuts in combination with very large amounts of negative emissions to stagnate and then

reverse the warming, creating the temperature overshoot outline. In SSP534-over, temperatures peak at 2.7°C above pre-industrial in 2077 after atmospheric $CO_2$ has reached its peak in 2062. The pathways are grouped into 3 phases:

• (I) the time until peak atmospheric $CO_2$ (2015-2062; 48 years), i.e., around net-zero $CO_2$ (+/- a few years (Koven et al., 2022)),

• (II) the time from peak atmospheric $CO_2$ until the end of SRM deployment (2063-2149; 87 years) and

• (III) the time after SRM deployment (2150-2249; 100 years).

With these phases, the initial phase of SRM until emissions get to net-zero and peak SRM deployment (I), the phasing out of
SRM and the reduction in atmospheric $CO_2$ concentration (II) and the dynamics after SRM stoppage (III) are captured. The simulations extend until 2249 but after 2200 land use change and GHG conditions are fixed and climate and carbon stores evolve without a change in forcing. SSP534-sulfur has all the baseline assumptions of SSP534-over but applies SAI on top to avoid crossing the 2°C-warming threshold. SAI is initialized in 2015 and deployment is carried on until 2150. SAI is represented in the simulation as a change in aerosol optical depth (AOD; Fig. 1b). The amount of AOD was determined with
a trial-and-error approach guided by the difference in energy balance between the SSP534-over scenario and the global mean radiative characteristics of a SSP126 scenario, which limits warming to 2°C. A curve was fit to the difference in energy balance between the two scenarios and this fitted difference in global mean forcing then translated into spatially resolved AOD using Tilmes et al.'s (2015) G4SSA AOD distribution. The use of the G4SSA AOD distribution is recommended by the GeoMIP protocol for models that cannot dynamically treat sulfur aerosols in the stratosphere (Kravitz et al., 2015) and has been done
with the CNRM-ESM model before (Baur et al., 2024a, b; Chen et al., 2023; Jones et al., 2022; Tilmes et al., 2022). See Visioni et al. (2021) for a comparison of the models participating in GeoMIP including CNRM-ESM2-1 with the prescribed G4SSA AOD distribution. G4SSA assumes equatorial injections (Tilmes et al., 2015). However, it has been demonstrated in other models that off-equatorial injection latitudes may perform better at compensating climate change impacts and reducing adverse side-effects from SAI (Kravitz et al., 2019; Tilmes et al., 2017; Visioni et al., 2023).

In this study, AOD was determined for the first ensemble member but applied to all three members equally. A sufficiently well calibrated SAI magnitude is classified as mostly staying in the range of 2°C +/-0.1°C of warming, as defined by us. Tilmes et al. (2020) use the CESM2-WACCM6 model configuration and a feedback-algorithm to determine SAI deployment magnitude

to also reduce temperatures from SSP534-over to 2°C. They require around half the magnitude of AOD than the SSP534-sulfur experiment in this study

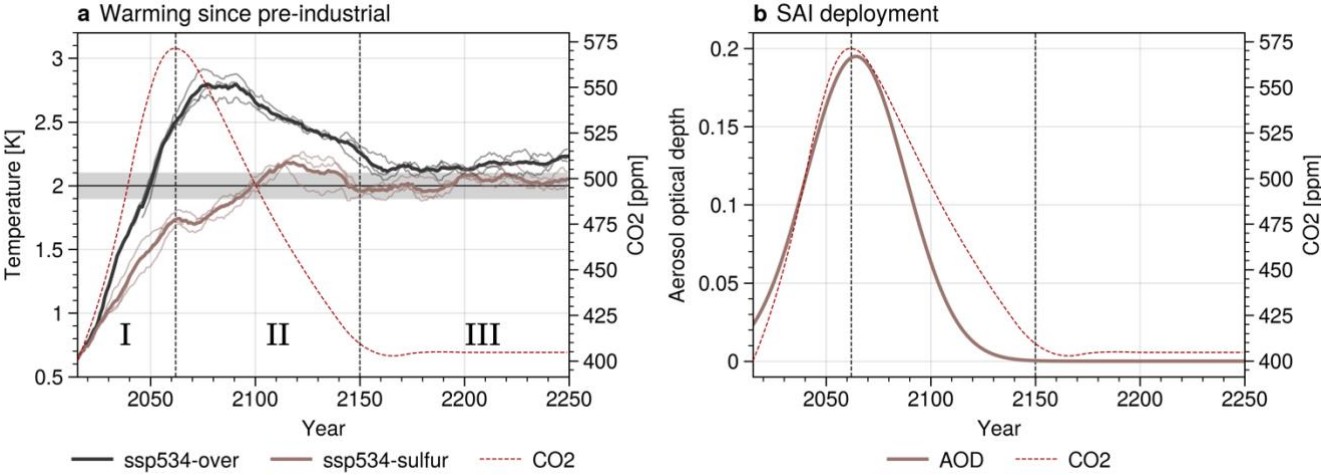

**Figure 1: a) 10-year rolling mean warming in the overshoot scenario SSP534-over (black) and the SAI peak-shaved scenario SSP534-sulfur (taupe). The gray zone indicates the 0.1°C tolerance level around the 2°C temperature target. Thick lines are the ensemble member means, thin lines the single members. The red stippled curve shows the atmospheric $CO_2$ concentration. B) Aerosol optical depth added in the SSP534-sulfur run as a proxy for SAI deployment. Stippled vertical lines indicate the overshoot phases I, II and III. Red stippled curve shows the atmospheric $CO_2$ concentration.**

The two experiments, SSP534-over and SSP534-sulfur, are run on the Earth System Model CNRM-ESM2-1+. CNRM-ESM2-1+ includes updates and improvements compared to the CNRM-ESM2-1 version used in CMIP6 (Fig. 2) (Séférian et al., 2019). Updated processes that impact the carbon cycle are the direct-diffuse light partitioning from aerosols which can affect the photosynthesis of plants, the crop harvesting which leads to a small reduction in the land carbon uptake, an improvement in water and carbon conservation in the soil due to land use and land cover change (LULCC) and an improved representation of the nitrogen fixation into the ocean which impacts the oceanic biological pump and leads to lower Net Primary Productivity (NPP) for an increase in global warming. The collective impact of these enhancements and updates is the improvement of the representation of the historical climate of the model and the modification of the transient climate response to cumulative emission (TCRE) of the model (Fig. 2). TCRE is slightly higher in CNRM-ESM2-1+ (1.76 °C per EgC) compared to the version used in CMIP6 (1.73 °C per EgC).

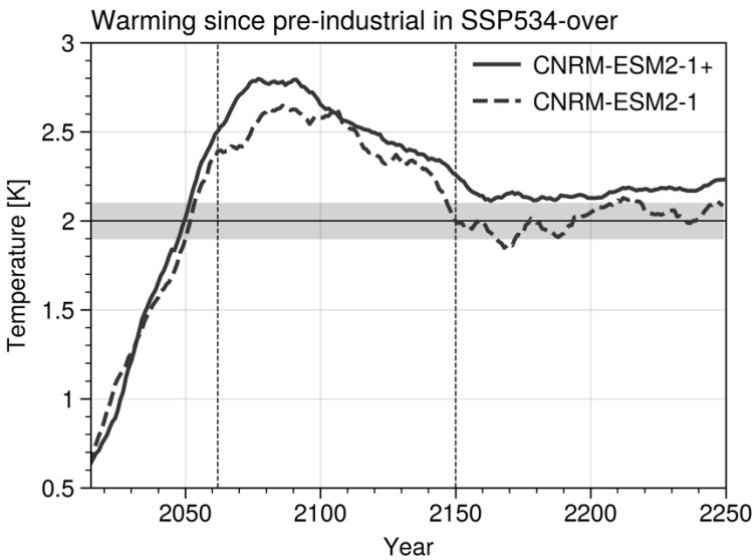

**Figure 2: 10-year rolling mean warming in the overshoot scenario SSP534-over simulated by the model version used in this study (CNRM-ESM2-1+; solid) and the former model version (CNRM-ESM2-1; dashed). Lines represent the mean of a 3-member ensemble. The gray zone indicates the 0.1°C tolerance level around the 2°C temperature target. Stippled vertical lines indicate the overshoot phases I, II and III.**

For each experiment, three realizations are performed with minimally perturbed atmospheric and oceanic parameterizations (perturbed at the 5th decimal). The results of this study are based on the mean of the three members except if indicated otherwise. Agreement on the carbon cycle processes is fairly consistent for the three members in both experiments. A 10-year rolling mean is displayed on the figures with the last 10 years of the historical runs of the same model version added to SSP534-over and -sulfur to be able to correctly calculate the rolling mean of the first 9 years of the experiments.

The simulations are run in concentration-driven mode as laid out by the CMIP6 SSP534-over representative concentration scenario guidelines (O'Neill et al., 2016). This means that the carbon cycle in our simulations reacts to this predetermined $CO_2$ concentration in the atmosphere and prespecified changes in land use and land cover but does not feed back to the atmospheric concentration of $CO_2$. In other words, any additional uptake or release by the carbon reservoirs will not be reflected in the atmospheric $CO_2$ and therefore global mean temperature. The prescribed $CO_2$ concentration facilitates the calculation of the amount of forcing required for the temperature reduction in the SSP534-sulfur run. However, to understand whether there is a difference in NEB between the two experiments, it is necessary to diagnose the corresponding anthropogenic $CO_2$ emissions

consistent with a given atmospheric $CO_2$ growth rate and a change in carbon uptake by land and ocean (see 2.2 Compatible

emissions).

## 2.2 Compatible emissions

The carbon flux from atmosphere to land and from atmosphere to ocean is calculated by the sub-models of CNRM-ESM2-1+: SURFEXv8.0 (Decharme et al., 2019; Delire et al., 2020) and NEMO3.6 (Mathiot et al., 2017). Taking the predetermined $CO_2$ concentration and the uptake by the carbon reservoirs into account, it is possible to infer how much $CO_2$ must have been

emitted to follow the prescribed atmospheric $CO_2$ concentration pathway. The carbon released by LULCC processes is not reflected in the $CO_2$ concentration and therefore the corresponding emissions are related to fossil fuel (FF) emissions only. The difference in the corresponding FF emission pathways between SSP534-over and SSP534-sulfur is used to indicate potential differences in NEB between an overshoot scenario and a peak-shaved scenario. The yearly compatible emissions are calculated in line with Friedlingstein et al. (2019), Jones et al. (2013), Koven et al. (2022) and Liddicoat et al. (2021) as:

$$E_{FF} = GCO_{2ATM} + S_{OCEAN} + S_{LAND} = GCO_{2ATM} + S_{OCEAN} + (NEP + E_{LULUCC}) \, , \qquad (1)$$

With $GCO_{2ATM}$ as the growth rate of atmospheric $CO_2$ in GtC per year, derived from the prescribed atmospheric $CO_2$ in parts per million (ppm) using the conversion of 1 ppm = 2.124 GtC (Ballantyne et al., 2015; Liddicoat et al., 2021). $S_{OCEAN}$ is the annual mean ocean carbon sink and $S_{LAND}$ the land sink (Net Biosphere Productivity, $NBP$), which is the Net Ecosystem Productivity ($NEP$) corrected for the disturbances from land-use change, harvest, grazing and fire ($E_{LULUCC}$). Additionally,

Gross Primary Productivity ($GPP$), the amount of carbon fixed during photosynthesis by all producers in the ecosystems, as well as the ecosystem physiological processes, Heterotrophic Respiration ($RH$) and Autotrophic Respiration ($RA$), i.e. the carbon released by soil ($RH$) and plants ($RA$), are examined.

## 3 Results

The compatible FF emission pathways show distinct features of an emission trajectory that leads to a temporary temperature

overshoot (Fig. 3a). Most of the first half of the 21[st] century is marked by a linear increase in emissions, which peak just before 2050 and then rapidly decline reaching net-zero around 2070 and max net-negative emissions by 2100. This maximum level in net-negative emissions is sustained for half a century until it is reduced to a smaller amount of net-negative emissions that is held constant until the end of the simulation. The CDR amount assumed in the Integrated Assessment Model REMIND-MAgPIE for SSP534-over is added as a dashed line to Fig. 3a, with the 2100 value extended for 50 more years for comparison

purposes. Figure 3b shows the difference between the compatible emission in Gt $CO_2$ per year (Fig. 3b): The first 50 years show a distinctly higher amount of compatible FF emissions under the SSP534-sulfur scenario than SSP534-over, which implies a reduced NEB. However, this effect is lost during 2075 to around 2150, where the difference between compatible

emissions is near-zero. After 2150, the end of SAI, allowable emissions under SSP534-over are slightly higher (Fig. 3b). In total, NEB is reduced by 60.4 Gt CO$_2$ (Fig. 3c,d). During Phase I, the additional uptake of 66.9 Gt CO$_2$ would imply a yearly

reduced NEB of 1.4 Gt CO$_2$. During Phase II, this amount gets reduced to 0.4 Gt CO$_2$ per year for the additional uptake of 31.1 Gt CO$_2$; and during Phase III the difference in emissions of -38 Gt CO$_2$ implies a 0.4 Gt CO$_2$ higher NEB per year in the SAI-scenario.

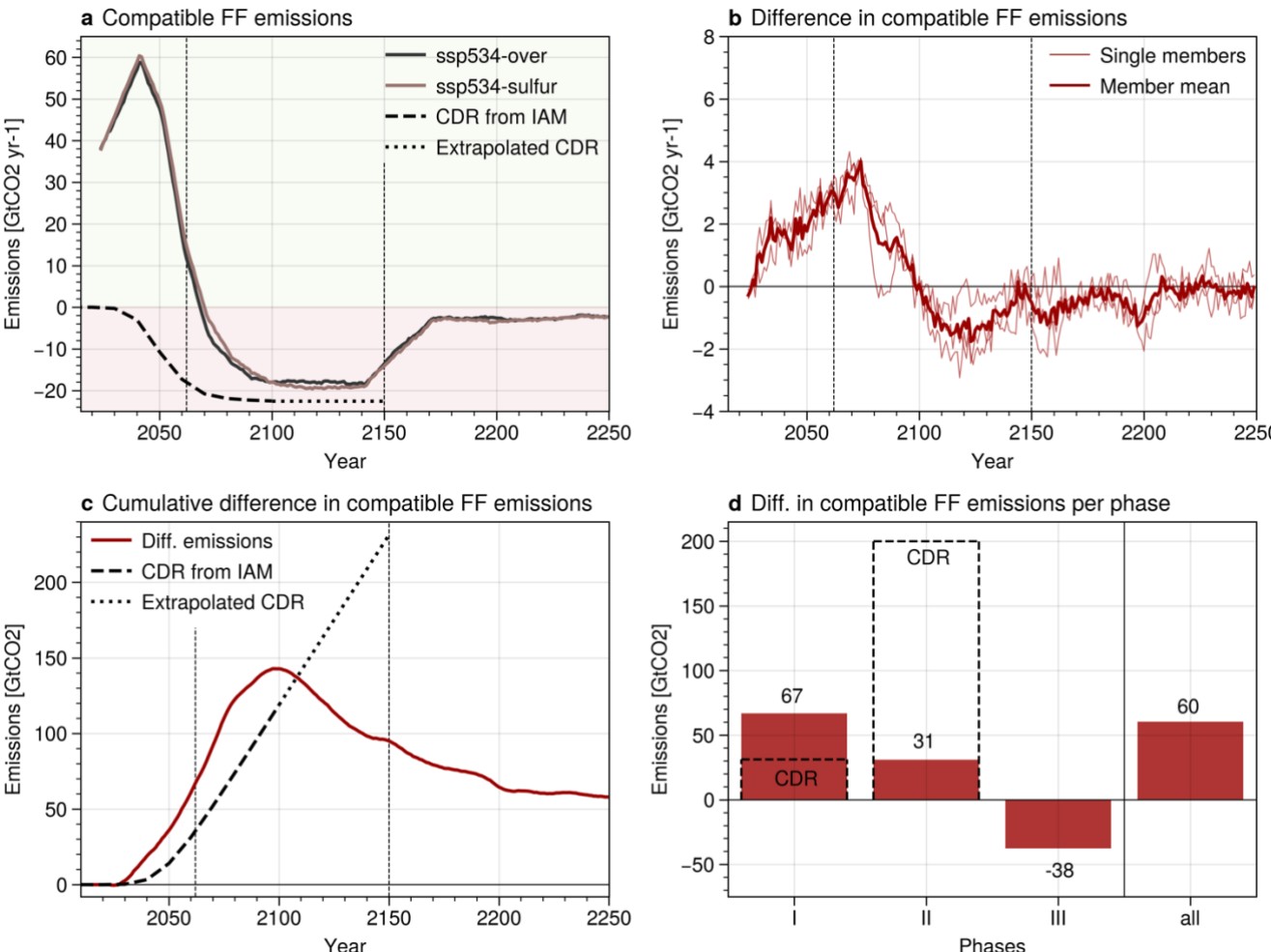

**Figure 3: a) compatible fossil fuel emissions for SSP534-over (black) and SSP534-sulfur (taupe). Dashed line shows the amount of**
**negative emissions implemented in the IAM ssp534-over scenario (Byers et al., 2022). b) SSP534-sulfur – SSP534-over. c) Shows**
**values of b in a cumulative manner. d) shows values of b when summed over the single peak-shaving phases (I, II and III) and**
**summed over the entire time frame (all). The stippled CDR-boxes indicate required CDR during the respective period. Stippled**
**vertical lines indicate the overshoot phases I, II and III. Data for these plots are based on the ensemble mean of the three members**
**except in b) where the single member differences are displayed in thin red lines.**


To better understand the processes behind this difference in compatible emissions between SSP534-over and SSP534-sulfur, Fig. 4 illustrates contrasts in carbon sink features between the two experiments. The annual difference in global ocean carbon uptake between the overshoot and peak-shaved scenario is small and most of the difference in annual global carbon uptake stems from the land sink (Fig. 4a). However, when the net cumulative uptake over the whole period is calculated, the size of the contribution to the total additional carbon uptake from land and ocean is not that different (Fig. 4b). This is because in the three different phases comprising the overshoot period, the ocean carbon uptake stays consistently slightly elevated while land carbon uptake varies between being enhanced and being reduced. On land, uptake is high in phase I, still elevated in phase II but low in phase III which leads to a total uptake that is similar to the total uptake of phase II.

Both ocean and land anthropogenic sinks become carbon sources during the 22nd century. While the ocean reverts back to being a small sink afterwards, land stays a source until the end of the experiment. It is very clear that for both ocean and land, pre-overshoot carbon uptake is not equal to the uptake post overshoot (Fig. 4a) at the same level of atmospheric $CO_2$ concentration (Fig. 4c) or amount of AOD (Fig. 4d). At least for the ocean, no trend is detectable in the timeframe of the simulation for the sink to develop back to its previous scale (Fig. 4a).

Panels e and f in Fig. 4 demonstrate that the additional carbon uptake under SSP534-sulfur remains even decades after the SAI deployment. The additional uptake in the ocean under SAI happens during the second half of the 21st century and remains equal to the annual uptake of SSP534-over uptake after that (Fig. 4f). Cumulative land carbon sink is maximized in 2100 for both experiments but the prior increase is higher under SSP534-sulfur and the subsequent rate of reduction is also higher than in SSP534-over (Fig. 4e).

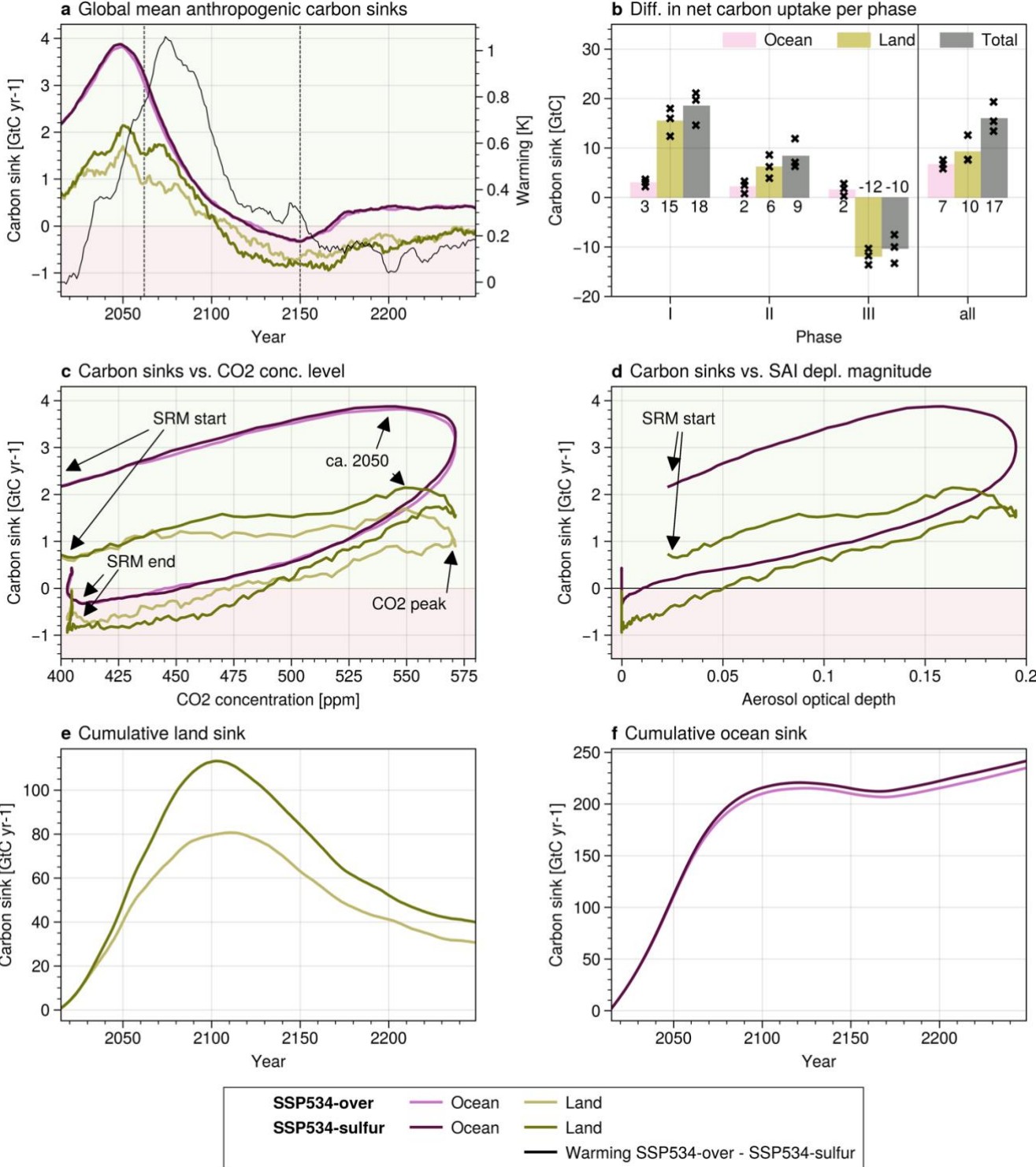

Figure 4: Carbon sink diagnostics. a) Global mean anthropogenic land and ocean carbon sinks for SSP534-over (light colors) and SSP534-sulfur (dark colors). Thin black line shows warming difference of SSP534-over – SSP534-sulfur. b) Difference (SSP534-sulfur – SSP534-over) in carbon uptake summed over the single peak-shaving phases (I, II and III) and summed over the entire time frame (all). Bars represent member mean, crosses single member results. c) Global mean anthropogenic carbon sinks versus global mean prescribed $CO_2$ concentration, d) SSP534-sulfur global mean anthropogenic carbon sinks versus global mean aerosol optical depth from SAI, e) cumulative global mean anthropogenic land carbon sink, f) cumulative global mean anthropogenic ocean carbon sink. Stippled vertical lines indicate the overshoot phases I, II and III. Data for these plots are based on the ensemble mean of the three members and are displayed as a 10-year rolling average.

During the first 100 years of the experiments, differences in $NEP$ are noticeable, as shown in Fig. 5a. Some of these differences are offset by the higher carbon flux from disturbances under SSP534-sulfur (Fig. 5b, c) when considering the total land sink (Fig. 4a). This may be due to the higher carbon density in the land carbon stores that, when burned or otherwise disturbed, release more carbon. $GPP$, $RA$ and $RH$ are higher under SSP534-sulfur than SSP534-over during most of the simulation (Fig. 5e). However, while $GPP$ increases rapidly after SAI deployment, $RA$ and $RH$ under SSP534-sulfur only diverge from the overshoot scenario after around 50 years of SAI (Fig. 5e,f) where an increase in these features (decrease in terms of carbon sink) offsets some of the increase in $GPP$. This might explain the substantial rise in carbon uptake during the first century, but a decrease in the difference of carbon uptake between the scenarios thereafter (Fig. 5a, 4a). $RA$ follows $GPP$ closely since photosynthesis drives the plant respiration which is followed by carbon storage in the soil.

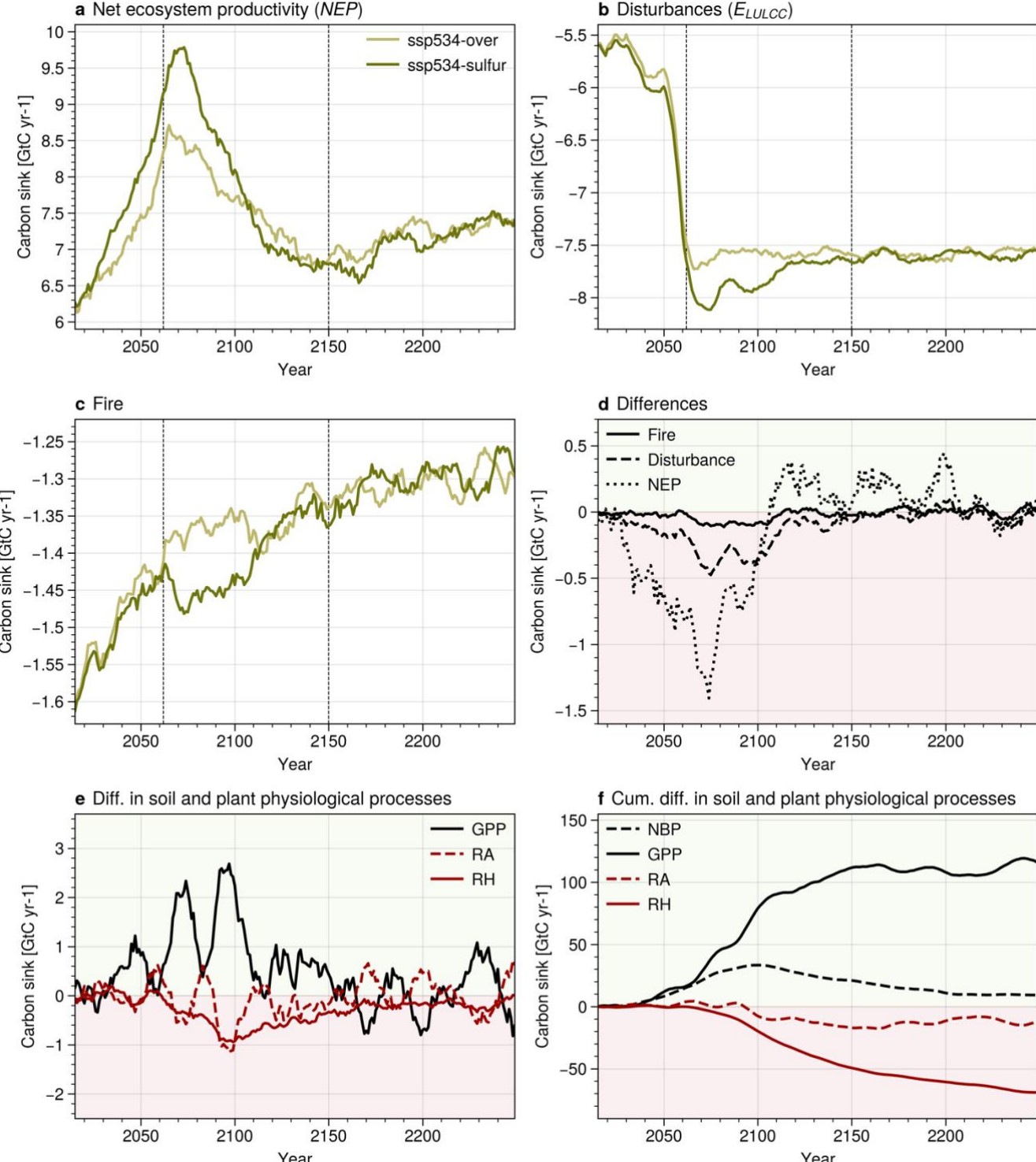

Figure 5: Land carbon sink diagnostics with a) Net Ecosystem Productivity (NEP), b) ecosystem disturbances, c) fire and d) difference (SSP534-sulfur – SSP534-over) in Net Primary Productivity (NEP), disturbances and fire, e) difference (SSP534-sulfur – SSP534-over) in Gross Primary Productivity (GPP), Heterotrophic Respiration (RH) and Autotrophic Respiration (RA) and f) like e) but with Net Biome Productivity (NBP) and cumulative difference. Data for these plots are based on the ensemble mean of the three members.

## 4 Discussion

We use CNRM-ESM2-1+ to simulate the global carbon cycle response in an overshoot versus an SAI peak-shaving scenario to determine differences in NEB between the scenarios if the same $CO_2$ concentration pathway is to be followed. This is a contribution to the discussion on the degree to which SAI could change underlying carbon dynamics in peak-shaving scenarios due to physical coupling with mitigation from carbon sink enhancement or degradation. The largest difference between SSP534-over and SSP534-sulfur in terms of emissions is seen in the first 50 years of SAI deployment (phase I) where SSP534-sulfur would require 67 Gt $CO_2$ fewer negative emissions than SSP534-over to follow the same atmospheric $CO_2$ concentration trajectory, which equates to around 2 years' worth of current annual anthropogenic emissions. During the phase-out of SRM (phase II), this carbon benefit gets reduced to 31 Gt $CO_2$ and switches to become a net disadvantage over phase III with -38 Gt $CO_2$.

Plazzotta et al. (2019) used a similar framework as employed in this study to determine additional "allowable emissions" due to carbon uptake benefits from SRM. They estimate, using output from six different Earth System Models running the GeoMIP G4 experiment, that around 147 Gt $CO_2$ additional emissions are "allowed" under SAI during the first 50 years of deployment due to carbon cycle benefits. At the same time, they suggest that around 50% of the additional carbon stored during the 50-year SAI intervention is released back to the atmosphere in the 50 years after a sudden termination of SAI (Plazzotta et al., 2019); Hence, their call for caution when comparing additional $CO_2$ uptake under SAI with CDR methods that store captured carbon in geological formations as the permanence and sustainability of geological carbon storage is not given for carbon sink enhancement under SRM. The G4 experiment is not comparable to the SSP534-sulfur simulation in this study and a sudden cessation of SRM is likely to cause different post-SRM impacts than a slow phase-out (Trisos et al., 2018). Nevertheless, the present study also suggests that some of the benefits in the terrestrial carbon uptake during SRM deployment in phase I are offset in the decades after the deployment (phase III) (Fig. 4b), highlighting the transient nature of SRM carbon sink enhancement–- even in scenarios without rapid termination.

Summed over the whole timeframe of our simulations, both ocean and land carbon uptake are enhanced under SAI (Fig. 4b). The clear difference in compatible emissions in Phase I is mainly due to modified terrestrial carbon cycle processes under SAI, rather than a change in marine carbon uptake (Fig. 4b). This dominance of the terrestrial carbon signal under SAI has also been documented by previous studies (Plazzotta et al., 2019). However, Tjiputra et al. (2016) contradict these results with a dominant ocean carbon uptake which they attribute to the strong nitrogen limitation on land in the model they use. Their scenario setup and model configuration differ from the one employed here in a way that they use prognostic atmospheric $CO_2$ and a pathway

that uses SRM to compensate for much more warming than SSP534-sulfur does. More recent studies have also found only minimal changes of land carbon uptake under SAI (Duan et al., 2020; Yang et al., 2020). These papers have only looked at what is considered Phase I of the peak-shaving deployment in this analysis. When all phases of the peak-shaving are taken into consideration, the net contribution of the ocean to the total carbon uptake under SAI is still clearly lower than that of land but makes up around 3/5 to 4/5 of total carbon uptake (Fig. 4b). This is because the land reacts more rapidly and more intensely to a change in forcing (large increase in carbon uptake in Phase I but also substantial decrease in phase III), while the ocean shows a small increase in Phase I that is not offset during later phases by a decrease (Fig. 4a,e,f). Similarly, Plazzotta et al. (2019) demonstrate how most of the carbon release after cessation comes from the land storage, while the sign of the ocean response is less pronounced and varies between the models.

In both experiments, land and ocean sink show a hysteresis-like behavior as a function of atmospheric $CO_2$ concentrations (Fig. 4c,d) where bringing atmospheric $CO_2$ down to pre-overshoot values does not restore carbon cycle dynamics to their pre-overshoot state. This could be due to a time lag between the atmospheric $CO_2$ and the recovery of the carbon sinks and is not unique to peak-shaving SRM conditions but a characteristic of atmospheric $CO_2$ overshoots. Hysteresis-like behavior has been found for several key climate variables in overshoot scenarios (Lee et al., 2021) and Fig. 4c shows how peak-shaving SRM cannot offset this behavior in terms of land and ocean carbon uptake. Figure 4a points to a relatively steady ocean uptake in the last 50 years of the two experiments which may imply either very slow recovery to the pre-overshoot state or, instead, a new stable state. In the terrestrial carbon uptake, even though forcing is unchanged in the last 50 years of the simulation, the land surface moves away from being a carbon source and reaches a balance between source and sink at the end of the experiment (Fig. 4a). The post-overshoot carbon cycle uptake may not have the same magnitude as pre-overshoot uptake since atmospheric $CO_2$ is kept stable whereas pre-overshoot $CO_2$ concentration was increasing.

A spatially resolved analysis may be able to explain the hysteresis-like behavior, since the global fluxes presented in this study cannot reflect regional differences in plant physiological processes and soil conservation. Future analyses should compare regional carbon uptake patterns before and after an overshoot for the same global mean temperature and atmospheric $CO_2$. Such a more refined regional analysis of the effect of SAI could additionally identify potential implications of SAI on specific land uses and land covers such as its impacts on food security and bioenergy for emission reduction purposes. This would add to the existing literature on the impact of SRM on specific crop types (Clark et al., 2023; Egbebiyi et al., 2024; Fan, 2023; Fan et al., 2021; Pongratz et al., 2012; Proctor, 2021; Proctor et al., 2018; Xia et al., 2014), which until now is focused on phase I of SRM deployment.

Several studies have demonstrated that SRM can substantially enhance *GPP* (e.g. Xia et al., 2016; Plazzotta et al., 2019; Yang et al., 2020). The main mechanism behind this enhancement differs between the comparison baseline. When compared to a mitigated climate, $CO_2$ fertilization seems to be the primary factor leading to an enhanced *GPP*, such as in Yang et al. (2020), Duan et al. (2020), Glienke et al. (2015), Govindasamy et al. (2002), Kalidindi et al. (2015) and Tilmes et al. (2020). However, when compared to a baseline with the same $CO_2$ concentration, reduced temperatures (Jin and Cao, 2023; Tilmes et al., 2020;

Tjiputra et al., 2016), the diffuse light fertilization (Xia et al., 2016) and SRM-induced hydrological changes (Muri et al., 2015, 2018; Tjiputra et al., 2016) can play a major role.

Another important factor in the magnitude of the terrestrial carbon cycle signal under SRM seems to be the nitrogen limitation
imposed in the model which can lead to very different results in terms of $GPP$ and $NPP$ (Tjiputra et al., 2016; Xia et al., 2016). Without a nitrogen limitation, the model overestimates the $CO_2$ fertilization effect on land vegetation and with that the land carbon uptake. In CNRM-ESM2-1+, $CO_2$ uptake on land is downregulated with a nitrogen limitation parameterization, whereby the land sink becomes less efficient with increasing $CO_2$ concentration. This may be one explanation as to why more recent studies find only a minor change between $NPP$ under SAI versus the same $CO_2$ concentration baseline without SAI,
such as Tilmes et al. (2020) and Duan et al. (2020) or even a decrease in $GPP$ and $NPP$ (Yang et al., 2020). These three studies, Tilmes et al. (2020), Duan et al. (2020) and Yang et al. (2020), are however based on different versions of the same model (CESM1 or CESM2 with the atmospheric component CAM4 or WACCM6), which might be an explanation for the similarity of the results.

Despite the decreased plant productivity indexes under SAI in Yang et al. (2020), the net terrestrial carbon uptake is still higher
than under the baseline when soil and plant respiration are taken into account (Yang et al., 2020). In contrast, the results of this study suggest that atmospheric carbon input from soil and plant respiration is enhanced under SSP534-sulfur compared to SSP534-over and experiences an augmented total land sink from SAI due to the large increase in $GPP$ (Fig. 5) rather than a decrease in respiration as in Yang et al. (2020). The results of this study show a larger difference in soil respiration between SSP534-sulfur and SSP534-over than in plant respiration. This may be attributable to the larger carbon storage in the soil under
SSP534-sulfur due to increased $GPP$ and hence the subsequent enhanced release of carbon from the soil. Also contrary to Yang et al. (2020), this study finds additional carbon release from disturbances under SSP534-sulfur during SAI deployment (Fig. 5b,c,d). Similarly, this may also be attributable to the larger amount of carbon that is stored by land and vegetation under SSP534-sulfur than SSP534-over and hence the larger fraction released when disturbed by harvest or fire. These increased disturbance carbon losses are likely highly model dependent and more studies analyzing these processes in detail are needed
to narrow down uncertainty related to a potential "carbon hangover".

A net enhancement in carbon uptake when summed over all three phases of the peak-shaving SAI deployment is calculated. The total carbon benefit (60 Gt $CO_2$) translates into 0.3 Gt $CO_2$ of annual CDR over the whole time period of 235 years. However, with 1.4 Gt $CO_2$ per year during the first almost 50 years (67 Gt $CO_2$ total), 0.4 Gt $CO_2$ during the following 87
years until SRM stoppage (31 Gt $CO_2$ total) and -0.4 Gt $CO_2$ during the last 100 years until the end of the experiments (-38 Gt $CO_2$ total). Compared to the annual negative emissions assumed in the underlying SSP534 scenario, these additional benefits and burdens appear to be of minor importance. However, these are non-negligible amounts considering the effort required to scale up negative emissions via CDR. For example, current estimates of total annual mitigation potential by 2050 are at 0.5-7 Gt $CO_2$/yr for afforestation and reforestation, 0.5-5 Gt $CO_2$/yr for BECCS and 2-4 Gt $CO_2$/yr for enhanced weathering (Beerling
et al., 2020; Dowling and Venki, 2018; Fuss et al., 2018). In fact, SRM has previously been referred to as a form of CDR

measure (Eliseev, 2012; Keith et al., 2017). However, scholars have emphasized that the net increase in $CO_2$ uptake under SRM is insufficient and unsustainable and cannot be considered as such (Muri et al., 2018; Plazzotta et al., 2019; Tjiputra et al., 2016). Given the variability of the terrestrial carbon fluxes in this study and the storage safety and storage timescales considered in common CDR technologies, this study supports the statement that the carbon cycle enhancement during peak-

shaving SAI is volatile and transient and cannot be referred to as CDR and counted as such. Nevertheless, the substantial reduction in annual NEB during the first few decades of the SSP534-sulfur experiment (phase I) supports the thought experiment of using SRM as a means to buy time for mitigation measures to take effect. However, it should be taken into account that during SAI phase-out (phase II), the NEB benefit is reduced and in phase III NEB is higher under SSP534-sulfur than -over and is a burden rather than a benefit. During periods of $CO_2$ concentration reduction (Phase II), the land and ocean

reservoirs turn into carbon sources rather than sinks for both SSP534-sulfur and -over, which means more CDR for the same $CO_2$ concentration reduction as before. In terms of carbon cycle processes alone, this may make it more difficult to reduce $CO_2$ concentration and phase out SRM, as the benefits of SRM and high $CO_2$ concentration lead to a very potent carbon-absorbing ecosystem.

It is evident from Figure 3b that there are two distinct periods for carbon uptake differences under SAI. The first period extends

until 2100, during which an additional 143 Gt $CO_2$ are taken up by terrestrial and marine reservoirs under SAI. The subsequent period goes until around 2220 where 83 Gt $CO_2$ less are taken up under SAI than the overshoot scenario. This transition from enhanced uptake to additional outgassing around the year 2100 corresponds temporally with the shift from land carbon sink to land carbon source under SSP534-sulfur and -over (Fig. 4a). This observation suggests that SAI does not cause the transition from sink to source, but rather reinforces the signal, enhancing uptake compared to the no-SAI scenario when it acts as a sink,

and enhancing outgassing when it acts as a source. As the transition from source to sink occurs at similar points in time in both scenarios, the underlying atmospheric CO2 concentration clearly plays a dominant role in the sink dynamics. However, the timing of the peak phase down of SAI is coincident with the shift from natural sink to source in SSP534-over, so the dynamical evolution of SAI in this experiment potentially causes two effects which cannot easily be disambiguated using this single experiment: SAI could potentially be enhancing the CO2-driven sink dynamics, but also the phase-out of SAI may be itself

causing a shift from carbon sink to source. Disambiguating these factors requires additional scenarios, such as a configuration that removes the negative emission phase while maintaining the same SAI deployment.

It should be noted that the results of this study are limited to one Earth System Model. As previous studies have found, carbon cycle processes can vary substantially between different models and increased robustness of the results could be achieved by

larger multi-model studies (Plazzotta et al., 2019). Especially models which are able to simulate sulfur injections in the stratosphere rather than using an offline-calculated AOD distribution would be a valuable addition to the literature. Injection design, such as injection timing and varying levels of injection latitude and altitude, has been found to have a significant impact on climate variables at the surface. It can therefore be assumed that different injection strategies would impact the carbon cycle to different degrees. Since the results of this study are constrained to equatorial injections, which have been found to be

suboptimal in terms of minimizing adverse side-effects from SAI deployment (Kravitz et al., 2019; Tilmes et al., 2017; Visioni et al., 2023), different injection strategies may lead to different results. Additionally, the 2°C temperature goal could have been met with different injection timing, such as a later start with a more sudden ramp-up of SAI, which could also result in different impacts on the carbon cycle. Furthermore, a wider range of underlying $CO_2$ concentration pathways should be analyzed, since larger or smaller overshoots, more or less, longer or shorter SAI deployment could affect carbon cycle processes and hence

the NEB result. SSP534-over was chosen as a baseline because it allows simulating an entire overshoot trajectory in less than 250 years. However, to achieve this, the scenario assumes large amounts of CDR already early in the present century and reaches the upper limit of currently estimated CDR capacity towards 2100 (Smith et al., 2023). Without the ability to perform such large-scale carbon removal, the temperature peak may be higher and the phase-out period substantially longer (Baur et al., 2023).

Recently, there has been a growing call for emission-driven climate simulations (Sanderson et al., 2023), rather than the concentration-driven approach taken in this study. This would increase the difficulty in determining the necessary SAI forcing and generating the SAI simulation since a temperature-carbon-cycle-feedback algorithm would need to be adopted, but could improve accuracy of the results as the compatible emissions framework could be omitted. Regardless, the SSP534-over compatible emissions trajectory determined in this study is in range with the compatible emissions by other Earth System

Models and the prescribed emissions from Integrated Assessment Models (Koven et al., 2022). Koven et al. (2022) used a former version of the CNRM-ESM2-1+ model, CNRM-ESM2. The difference between the pathways in their study and the present one is attributable to the updates made to the model that affect the carbon cycle response (see 2.1 Model and Simulations). Lastly, this study looked at one type of SRM. For a more complete picture on NEB differences under SRM, future analyses should consider other studied SRM approaches such as Marine Cloud Brightening and Cirrus Cloud Thinning

as well, which have been shown to have differing impacts on the carbon cycle (Duan et al., 2020; Lauvset et al., 2017; Lee et al., 2020; Muri et al., 2015, 2018).

**5 Conclusion**

In this study, Negative Emission Burden (NEB) is compared between an overshoot scenario (SSP534-over) and a peak-shaving pathway (SSP534-sulfur) from 2015 to 2249. In the peak-shaving pathway, SAI is used to reduce temperatures to 2°C of

warming compared to pre-industrial, instead of peaking at 2.7°C as in the overshoot case. For this purpose, SAI deployment starts in 2015, reaches its peak in 2070 and is terminated in 2150. The atmospheric $CO_2$ concentration in both experiments is prescribed by the CMIP6 guidelines (O'Neill et al., 2016). Hence changes in atmospheric $CO_2$ due to carbon cycle variations are not represented but a framework laid out in previous studies (Friedlingstein et al., 2019; Koven et al., 2022; Liddicoat et al., 2021) is used to determine the amount of fossil fuel emissions compatible with the prescribed $CO_2$ concentration when

additional uptake or release by the marine and terrestrial carbon reservoirs are taken into account.

This study finds that NEB is 60 Gt $CO_2$ lower under SAI compared to the overshoot scenario when summed over the whole timeframe of the trajectory (235 years), but benefits are skewed towards the early years of SRM deployment. NEB is reduced during the first few decades of SAI deployment until net-zero $CO_2$ by 67 Gt $CO_2$. During this phase, both land and ocean carbon sinks give extra negative emissions worth around 1.4 Gt $CO_2$ of annual CDR. During the phase-out of SAI, NEB is still

enhanced but reduced to an annual benefit of 0.4 Gt $CO_2$ and turns into a burden of additional NEB after SAI termination of 0.4 Gt $CO_2$ additional annual CDR mostly due to soil carbon respiration. Overall, around two thirds of the carbon uptake benefit under SAI come from the terrestrial land sink and a third comes from the ocean. The land sink is more dynamic to changes in SAI, as uptake is substantially increased during SAI roll-out, but reduced during parts of SAI phase-out and post-deployment, whereas ocean sink is slightly enhanced during the roll-out period but stays close to the overshoot baseline

thereafter.

The reduction in annual NEB during the first few decades of the SSP534-sulfur experiment confirms the idea of using SRM as a means of buying time since CDR burden is reduced. But benefits are largely restricted to the early phase of deployment, with reduced benefits during SAI ramp down and enhanced carbon release from disturbance post deployment. The additional challenge in reducing atmospheric $CO_2$ concentration during the subsequent phase of the peak-shaving scenario may make

SAI phase-out difficult and undesirable. Multi-model studies looking at a greater variety of peak-shaving pathways are needed to confirm the results of this study.

**Code and data availability statement**

The code is available at https://doi.org/10.5281/zenodo.14753918 (Baur, 2025).

**Author Contribution**

All authors conceptualized the study and SB carried it out. SB ran the simulations and prepared the manuscript with contributions from all authors.

**Ethics Declaration**

One of the co-authors is on the editorial board of Earth System Dynamics. The authors declare no other conflicts of interest.

**Acknowledgements**

Susanne Baur is supported by CERFACS through the project MIRAGE. BS and RS acknowledges funding by the European Union's Horizon 2020 (H2020) research and innovation program under Grant Agreement No. 101003536 (ESM2025 – Earth System Models for the Future), 821003 (4C, Climate-Carbon Interactions in the Coming Century) and 101003687 (PROVIDE).

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
