# Peer review of "Change in negative emission burden between an overshoot versus peak-shaved Stratospheric Aerosol Injections pathway"

_EGUsphere, 2024_

## Referee Comment (RC1)

**Peer review of "Change in negative emission burden between an overshoot versus peak-shaved Stratospheric Aerosol Injections pathway"**

Journal: Earth System Dynamics

Authors: Susanne Baur, Benjamin M. Sanderson, Roland Séférian, Laurent Terray

Article posted: 5 September 2024

Report submitted: 9 October 2024

**General Comments**

This study examines stratospheric aerosol injection (SAI) in the CNRM-ESM2-1+ climate model. Specifically, the study considers the SSP5-3.4 "overshoot" scenario with and without an idealized SAI strategy which limits global warming to approximately $2°C$. The authors look at carbon fluxes between the atmosphere and the land and ocean components; because the $CO_2$ concentrations in the atmosphere are fixed throughout, any change in flux to or from the land or ocean must be balanced by a change in anthropogenic emissions. Therefore, changes to these fluxes represent how much additional (or less) anthropogenic emissions are "allowed" in order to follow a given pathway under a given intervention. The authors conclude that, in this experiment, the SAI intervention allows for approximately 60 Gt of additional $CO_2$ emission during SAI ramp-up, 30 Gt during peak deployment, and -30 Gt during phase-out.

The study will be suitable for publication with little additional work. The writing is very high-quality; the text is extremely clear, citations in the introduction and discussion are plentiful, and comparisons to other studies are thorough. I also think the paper identifies an important gap in the literature, as most SAI simulations only run for 30-50 years or until the end of the century, while this study follows SAI through peak deployment, wind-down, termination, and post-deployment out to 2250. However, I would like to see more documentation for the study's methodology, including the process of copying aerosol optical depth (AOD) fields from one model, scaling them up or down, and pasting them into another model, and more acknowledgement that the results may be influenced by the methodology and SAI strategy chosen. I offer specific comments below.

**Specific Comments**

Abstract, line 19-20: "SAI is used to maintain $1.5°C$ warming" - I think this is a typo; the rest of the article describes SAI being used to limit warming to $2°C$.

Abstract, lines 20-23 and generally throughout: this study only examines one SAI strategy. While this is perfectly fine (many studies only look at one strategy), the summarized results in the last sentence of the abstract *may* not be applicable to other strategies - e.g., if the initial injection

rate were constant instead of ramped up, if the injection happened at different locations, or if the total amount of cooling were different.

Line 100, "a global mean temperature increase of 2℃" - relative to what? Temperatures in this study are all presented as an increase relative to some baseline (presumably some preindustrial value), but it is never stated clearly what that baseline is or how it is computed. The baseline value, and its definition, should be stated clearly.

Lines 116-117, "The amount of AOD was determined with a trial-and-error approach…" - more needs to be said about how the "injection rates" (AOD magnitude) were chosen. The text offers the explanation that AOD was chosen to limit warming to 2℃ ± 0.1℃, but injection starts well before 2℃ is reached, and all three ensemble members appear to overshoot the tolerance level around ~2115. The curve in Fig. 1b is very smooth, suggesting that a desired AOD curve was fit, and the shape of the curve tweaked rather than the AOD in each individual year (is 1b a 10-year running mean?). Was AOD chosen first for one ensemble member, and then the same quantity used for the other two ensemble members? This is important because the main message of the paper is "SAI which limits warming to 2℃ does X and Y to the carbon cycle," but the magnitude of AOD used doesn't seem to have been chosen purely based on the 2℃ target.

Lines 118-119, "The difference in global mean forcing was then translated into spatially resolved AOD using Tilmes et al.'s (2015) G4SSA AOD distribution" - this needs much more explanation. Tilmes, et al. (2015) simulated 8 Tg $SO_2$ injection over the equator during the years 2020-2070 and presented the AOD distribution for that injection strategy. It sounds like this study copied and pasted that AOD distribution into this model, and then scaled it up or down. This is a substantial idealization, and needs to be discussed more - have other models or studies used the AOD fields provided by Tilmes, et al. (2015)? Has it been done in this model before? Has this model done simulations of SAI or volcanic eruptions? The authors should also mention that this AOD distribution corresponds to equatorial SAI, which is relevant because a.) injections at different latitudes could have different results, and b.) equatorial SAI is less commonly studied now because it is known that the aerosols tend to remain confined to the tropical pipe, over-cooling the tropics and under-cooling the poles.

Lines 119-120, "A sufficiently well calibrated SAI magnitude is classified as mostly staying in the range of 2℃ ± 0.1℃ of warming" - is this your definition, or someone else's?

Figure 1: it would be very helpful to add atmospheric $CO_2$ concentrations to this figure somehow, either as a separate panel or as a right-hand-side axis on panel (a). They're an integral part of the study's methodology and also play a role in how the three "phases" are defined, but don't appear to be shown anywhere.

Lines 144-145, "the results of this study are based on the ensemble mean of the three members except if indicated otherwise" - this should be stated everywhere it is relevant (e.g., in each figure caption) rather than hidden in one place in the text.

Lines 145-146, "a 10-year rolling mean is displayed on the figures" - is this true for all panels of all figures? If so, this should be stated in each figure caption, rather than buried in the text.

Figure 4b: I recommend adding some kind of errorbar or ensemble spread to this panel.

Figures 4c-4d: Not strictly necessary, but it would be helpful to the reader to add markers to these panels to denote different points in time - it is not immediately obvious to the reader which end of each curve represents the start of the experiment, what year the "peak" represents, and so on.

Discussion, last paragraph: The authors do a good job of discussing the limitations of using one model and one emissions scenario, but I recommend they also address a.) the idealized nature of the experiment (using prescribed AOD instead of simulating aerosols) and b.) that only one SAI strategy was considered. The same evolution of global mean AOD could have been accomplished with different AOD distributions representing, for example, subtropical or subpolar injection instead of equatorial. Additionally, the 2℃ temperature goal could also have been met with other amounts of AOD; for example, in SSP5-3.4-over, the 2℃ threshold is reached in ~2050 without any SAI - consider an experiment where you used same emissions scenario but with no injection until 2050, at which point SAI is ramped up very suddenly to maintain the 2℃ target. Such a change would probably affect this study's conclusions substantially, while still claiming similar methodology.

---

## Author Comment (AC1)

**Peer review of "Change in negative emission burden between an overshoot versus peak-shaved Stratospheric Aerosol Injections pathway"**

Journal: Earth System Dynamics
Authors: Susanne Baur, Benjamin M. Sanderson, Roland Séférian, Laurent Terray Article posted: 5 September 2024
Report submitted: 9 October 2024

General Comments

This study examines stratospheric aerosol injection (SAI) in the CNRM-ESM2-1+ climate model. Specifically, the study considers the SSP5-3.4 "overshoot" scenario with and without an idealized SAI strategy which limits global warming to approximately 2°C. The authors look at carbon fluxes between the atmosphere and the land and ocean components; because the $CO_2$ concentrations in the atmosphere are fixed throughout, any change in flux to or from the land or ocean must be balanced by a change in anthropogenic emissions. Therefore, changes to these fluxes represent how much additional (or less) anthropogenic emissions are "allowed" in order to follow a given pathway under a given intervention. The authors conclude that, in this experiment, the SAI intervention allows for approximately 60 Gt of additional $CO_2$ emission during SAI ramp-up, 30 Gt during peak deployment, and -30 Gt during phase-out.

The study will be suitable for publication with little additional work. The writing is very high-quality; the text is extremely clear, citations in the introduction and discussion are plentiful, and comparisons to other studies are thorough. I also think the paper identifies an important gap in the literature, as most SAI simulations only run for 30-50 years or until the end of the century, while this study follows SAI through peak deployment, wind-down, termination, and post-deployment out to 2250. However, I would like to see more documentation for the study's methodology, including the process of copying aerosol optical depth (AOD) fields from one model, scaling them up or down, and pasting them into another model, and more acknowledgement that the results may be influenced by the methodology and SAI strategy chosen. I offer specific comments below.

We thank the reviewer for taking the time to go through our work and the constructive feedback. We are pleased to hear that they find the study to be a valuable contribution to the SRM discourse and hope to address their concerns with our modifications.

Specific Comments
Abstract, line 19-20: "SAI is used to maintain 1.5°C warming" - I think this is a typo; the rest of the article describes SAI being used to limit warming to 2°C.

Yes, thank you.

Abstract, lines 20-23 and generally throughout: this study only examines one SAI strategy. While this is perfectly fine (many studies only look at one strategy), the summarized results in the last sentence of the abstract *may* not be applicable to other strategies - e.g., if the initial injection rate were constant instead of ramped up, if the injection happened at different locations, or if the total amount of cooling were different.

Yes, the paper only relates to SAI, hence SAI in the title and everywhere else in the text. We have a short mention in the discussion on the fact that results would likely differ for Marine Cloud Brightening or Cirrus Cloud Thinning. We have added more explanation related to the dependence of the results on the chosen SAI strategy and potential deviations for different deployment designs. We will add a sentence to the abstract (L.20-23) along the lines of "Results may differ for other injection strategies and deployment design".

Line 100, "a global mean temperature increase of 2°C" - relative to what? Temperatures in this study are all presented as an increase relative to some baseline (presumably some preindustrial value), but it is never stated clearly what that baseline is or how it is computed. The baseline value, and its definition, should be stated clearly.

We will add "Warming since pre-industrial" to the titles of Figure 1a and Figure 2 and add a sentence on our definition of temperature baseline in the methods.

Lines 116-117, "The amount of AOD was determined with a trial-and-error approach..." - more needs to be said about how the "injection rates" (AOD magnitude) were chosen. The text offers the explanation that AOD was chosen to limit warming to 2°C ± 0.1°C, but injection starts well before 2°C is reached, and all three ensemble members appear to overshoot the tolerance level around ~2115. The curve in Fig. 1b is very smooth, suggesting that a desired AOD curve was fit, and the shape of the curve tweaked rather than the AOD in each individual year (is 1b a 10-year running mean?). Was AOD chosen first for one ensemble member, and then the same quantity used for the other two ensemble members? This is important because the main message of the paper is "SAI which limits warming to 2°C does X and Y to the carbon cycle," but the magnitude of AOD used doesn't seem to have been chosen purely based on the 2°C target.

We are going to add clarifications on the design of the SAI scenario. In short: Indeed, the AOD curve was fit. 1b is not a 10-year running mean. The same AOD was used for all members, the only difference in the members is the background state of the SSP585 baseline ensemble members.

Lines 118-119, "The difference in global mean forcing was then translated into spatially resolved AOD using Tilmes et al.'s (2015) G4SSA AOD distribution" - this needs much more explanation. Tilmes, et al. (2015) simulated 8 Tg $SO_2$ injection over the equator during the years 2020-2070 and presented the AOD distribution for that injection strategy. It sounds like this study copied and pasted that AOD distribution into this model, and then scaled it up or down. This is a substantial idealization, and needs to be discussed more - have other models or studies used the AOD fields provided by Tilmes, et al. (2015)? Has it been done in this model before? Has this model done simulations of SAI or volcanic eruptions? The authors should also mention that this AOD distribution corresponds to equatorial SAI, which is relevant because a.) injections at different latitudes could have different results, and b.) equatorial SAI is less commonly studied now because it is known that the aerosols tend to remain confined to the tropical pipe, over-cooling the tropics and under-cooling the poles.

Yes indeed, the reviewer correctly identified the SAI modeling process using a prescribed AOD distribution. Using the G4SSA AOD distribution is recommended by the GeoMIP protocol for models that cannot dynamically treat sulfur aerosols in the stratosphere (Kravitz et al., 2015) and has been done with this model before (Jones et al. 2022, Chen et al. 2023, Tilmes et al. 2022, Baur et al., 2024a,b). Visioni et al., 2021 compares the models participating in GeoMIP including CNRM-ESM2-1 with the prescribed G4SSA AOD distribution.

The offline calculation of an AOD field is also done by other ESMs (MPI-ESM prescribes its aerosol distribution from simulations described in Niemeier and Schmidt (2017) and Niemeier et al. (2020)).

We are going to include more details on the SAI setup in the text, the latitude of injection in the AOD distribution and its potential impact on results.

Lines 119-120, "A sufficiently well calibrated SAI magnitude is classified as mostly staying in the range of 2°C ± 0.1°C of warming" - is this your definition, or someone else's?

We are going to add that this is our classification.

Figure 1: it would be very helpful to add atmospheric $CO_2$ concentrations to this figure somehow, either as a separate panel or as a right-hand-side axis on panel (a). They're an integral part of the study's methodology and also play a role in how the three "phases" are defined, but don't appear to be shown anywhere.

Yes, we agree with the reviewer and are going to add CO2 concentration as a second y-axis.

Lines 144-145, "the results of this study are based on the ensemble mean of the three members except if indicated otherwise" - this should be stated everywhere it is relevant (e.g., in each figure caption) rather than hidden in one place in the text.

We are going to add this information in every figure caption where relevant.

Lines 145-146, "a 10-year rolling mean is displayed on the figures" - is this true for all panels of all figures? If so, this should be stated in each figure caption, rather than buried in the text.

Same as above.

Figure 4b: I recommend adding some kind of errorbar or ensemble spread to this panel.

We will add the ensemble spread to the bars.

Figures 4c-4d: Not strictly necessary, but it would be helpful to the reader to add markers to these panels to denote different points in time - it is not immediately obvious to the reader which end of each curve represents the start of the experiment, what year the "peak" represents, and so on.

Yes, we will add some clarification to these two panels.

Discussion, last paragraph: The authors do a good job of discussing the limitations of using one model and one emissions scenario, but I recommend they also address a.) the idealized nature of the experiment (using prescribed AOD instead of simulating aerosols) and b.) that only one SAI strategy was considered. The same evolution of global mean AOD could have been accomplished with different AOD distributions representing, for example, subtropical or subpolar injection instead of equatorial. Additionally, the 2°C temperature goal could also have been met with other amounts of AOD; for example, in SSP5-3.4-over, the 2°C threshold is reached in ~2050 without any SAI - consider an experiment where you used same emissions scenario but with no injection until 2050, at which point SAI is ramped up very suddenly to maintain the 2°C target. Such a change would probably affect this study's conclusions substantially, while still claiming similar methodology.

Ok, we will add more scenario disclaimers to the discussion and emphasize that the results are influenced by the chosen methodology and SAI strategy.

---

## Author Comment (AC2)

**Reviewer #2**

**General Comments**

The paper examines the facilitation and inhibition of CO2 uptake throughout the entire duration of a peak-shaving deployment of SAI. A three-member ensemble simulation of SSP5-3.4-over is compared against SSP5-3.4-sulfur in CNRM-ESM2-1+. The paper finds finds that although there is an enhancement of the carbon cycle during the early phases of the deployment, the later phases see a decline in enhancement, eventually becoming an emission burden in the final phases of peak-shaving and in the years after deployment.

I believe that this paper has valuable contributions and makes some excellent points about the impermanence of SAI's carbon cycle enhancement in a peak-shaving scenario. However, there are portions of the paper in which the language can be imprecise, and the points made are overreaching. In particular, the paper states that phasing out SAI may be made less desirable or more difficult by the burden incurred in the later phases. This statement seems to make certain assumptions about the scenario. In particular, either it assumes that those in the scenario take advantage of the entirety of enhancement provided by SAI in the early stages and ignore that some of SAI's carbon benefit is temporary, or it assumes that those doing CDR in the scenario care more about smaller rate of carbon removal they can do in the later phases than the smaller amount of carbon removal they must do in the later phases.

In all, I believe the science of this paper to be very good, but it draws conclusions from its findings in a way that should be revised.

We thank the reviewer for their valuable comments and recommendation for publication if major points are addressed. We are going to modify the paper to exclude any statements regarding a scenario's desirability, since it is ultimately a value judgement and, as the reviewer points out, makes assumptions on the scenario. We are going to add more clarity on the temperature baseline, add details to plots, add an additional paragraph on the areas above and below 0 land and ocean carbon uptake and other little corrections and clarifications the reviewer kindly pointed out.

We hope that our modifications will address the reviewers' concerns and thank them for a second screening of the paper.

**Specific Comments**

1. Some notes around the concept of making phasing SRM out more difficult:
    1. Line 92: Prolonged SRM deployment and higher CDR in response to sink degradation are a sliding scale, not necessarily an "and". For

example, higher CDR to exactly offset the source in phase III of SSP534-sulfur would result in the same phase out, if I'm understanding correctly.

We are going to add an "or" to emphasize that a sink degradation would prolong SRM deployment **or** require larger amounts of CDR to compensate.

2. The statements (e.g. Line 382) that the phase-out of SAI may be made "undesirable" can be further developed, as certain assumptions about those desires can be be made explicit.

   1. If the global community uses up the extra $CO_2$ budget gained from the early phases of peak-shaving by either not mitigating or doing less CDR, then would they have the $CO_2$ concentrations of SSP534-over and thus an extra burden on the tail end. It is relatively clear how the people of Phases II and III might not want to ramp up CDR more than they would have had to.

   2. A less clear example is that if is if they mitigate and do CDR in Phase I as they would have in the baseline SSP534-over case, then they would have lower $CO_2$ concentrations in Phases II and III. They would need more CDR for the same $CO_2$ removal during these phases, but they would also have less $CO_2$ to remove (in net, 60 Gt less). If the speed at which carbon cycle draws out $CO_2$ is the primary desire, they would find difficulty, but if they value getting the $CO_2$ concentration back down to a certain level while maintaining a certain temperature (the premise of peak shaving), I struggle to see how phasing out SAI would be made more undesirable.

The reviewer lists some interesting scenario logics. Our thought process following the results is: Carbon uptake under SAI throughout the peak-shaving period varies in a way that SAI enhances the already pre-existing effect in overshoot scenarios where reducing atmospheric $CO_2$ concentration is "easier" before net-zero $CO_2$ and "harder" afterwards. Therefore, regardless of whether the additional $CO_2$ budget is taken advantage of in the early stage or not, the additional outgassing under SAI after net-zero makes atmospheric $CO_2$ reduction "harder" in later stages.

With SRM keeping temperatures at a comfortable level and CDR becoming harder, we concluded that this may make SRM phase-out less desirable than for example in an overshoot scenario where no peak-shaving is happening. We recognize that this is a subjective statement and are going to entirely remove the framing of "desirability" from the paper.

However, despite desirability, our conclusions are independent of whether the additional carbon uptake benefit is leveraged or not. Therefore, we refrain from discussing any potential scenario outlooks or assumptions regarding the "use" of the additional carbon budget.

Lines 324-328 / Figure 3: the extra 38 GT of $CO_2$ burden seems like it would be less than two years of extra CDR as modeled in ssp534-sulfur, ending in 2152 instead of 2150. While I agree with the "non-negligible" statement with respect to CDR that exists today, the CDR in the paper's scenarios far outpace the examples the paper gives. This should be at least addressed.

We are going to add a sentence in L. 328 along the lines of "However, compared to the annual CDR assumed in the scenario, these additional CDR efforts appear less substantial"

2. Line 21: The concept of "buying time" during peak shaving usually refers to time it takes halt temperature rise and reduce it again, and the risks incurred during that temperature peak. It does not usually refer to the extra help from carbon cycle enhancement.

    1. E.g. Zarnetske PL, Gurevitch J, Franklin J, Groffman PM, Harrison CS, Hellmann JJ, Hoffman FM, Kothari S, Robock A, Tilmes S, Visioni D, Wu J, Xia L, Yang CE. Potential ecological impacts of climate intervention by reflecting sunlight to cool Earth. Proc Natl Acad Sci U S A. 2021 Apr 13;118(15):e1921854118. doi: 10.1073/pnas.1921854118. PMID: 33876741; PMCID: PMC8053992.

The "buying time" concept has different interpretations with the most common one being to "reduce pressure" of implementing adaptation and mitigation.

*"Insofar as SAI combats the worst impacts of peak global $CO_2$ emissions, it buys time for different climate approaches that might take longer to show effect, such as mitigation, adaptation or other CDR measures. SAI, as a stop-gap measure with quick results, reduces pressure for implementing other adaptation and decarbonization strategies. It is this reducing pressure framing that seem to lie at the heart of the many versions of buying time or peak-shaving arguments." Neuber & Ott, 2020* (*https://www.mdpi.com/2076-3417/10/13/4637*)

3. The difference in compatible emissions do not seem linked to the three phases of SAI deployment specifically, considering the crossover at 2100. Looking at 3b and 4a (for land at least), it looks as though under SSP534-sulfur, the sink becomes more of a sink and the source becomes more of a source. To say in line 377 that the uptake is increased during SAI roll-out but decreased during phase-out feels off, since it seems to have more

to do with when it's a source or sink, and thus the background GHG emissions pathway and the CO2 concentrations.

1. In other words, would phasing out SAI while doing no CDR cause the negative difference starting at 2100 in Figure 3b? Alternatively, would not phasing out SAI and maintaining a constant amount of cooling while doing the same amount of CDR as simulated create the 2100 crossover?

We agree with the reviewer that the response of the land reservoir does not correlate directly with the stages of SAI deployment. We are going to change the framing of the sentence in L. 377 and throughout the paper to make clear that while the C uptake is enhanced when land acts as a sink, it is reduced when land acts as a source. However, the primary mechanism of whether it is a sink or driver depends on the CO2 concentration, not SAI.

4. Line 26: Although CO2 removal is often considered a form of mitigation (IPCC AR6 WGIII: CDR Factsheet), it is also often considered separate from mitigation (e.g. NASEM, 2021: "This portfolio must involve reducing GHG emissions to the atmosphere (mitigation), and removing carbon from the atmosphere and reliably sequestering it"). This paper should explicitly state whether it is defining mitigation to include CDR or considering CDR to be independent, cite a source to support that definition, and then stick to it for the duration of the paper.

1. I would recommend separating the two, such that something like the first line of the abstract "…allowing additional time for the implementation of conventional climate mitigation strategies and CO2 removal." could be read without confusion, regardless of whether the reader defines CO2 removal as mitigation or not.

We thank the reviewer for pointing this out and will add a sentence in the introduction clarifying our definition of what counts as mitigation. We are going to go with the denomination of the latest IPCC report, that classifies CDR as a form of mitigation (Riahi et al., 2022).

5. Lines 37-78: Be clear about the baseline that the paper is comparing SAI to with each comparison. For example, the CO2 fertilization effect is likely to the same temperature, no-SAI, but the reduced heat stress is likely to the same CO2 concentration or year, no-SAI.

We agree that this is important, as highlighted in L. 292, and are going to include the baselines to the comparisons.

6. Line 100: Regarding "increase of 2°C", say relative to what baseline (pre-industrial, but also discuss how are it is being defined)

We will add details on the definition of our temperature baseline.

7. Line 117: The paper should give SSP126 the same background as it gave SSP534. At present, it is put into the text without definition.

We disagree with the reviewer's judgment here since our scenarios have nothing to do with the SSP126 storyline itself and no comparisons of our simulations with a SSP126 pathway are made. We are going to add the words "the global mean radiative characteristics of a SSP126 pathway" in L 117 to add clarification.

8. Figure 3b: I am curious about the total area over 0 and under 0 (which sum to be 60 Gt), especially since there seem to be two "phases": where the difference is positive going until ~2100 and where it is negative after.

We are going to add a short paragraph in the Discussion where we quantify separately the area over and under 0, since they can be easily separated into 2 phases (before 2100, after 2100).

9. Figure 3b: It would be nice to have a measure of variability – what is significant difference between compatible emissions vs. what is a byproduct of variability.

Yes, agreed. We are going to add a measure of the spread around the mean difference to the plot.

10. Figure 4: Some sort of Carbon sink vs. Temperature plot may be useful here. Masking temperature by land and ocean could be included, or use cooling done by SAI vs. difference in sinks, but given that peak-shaving SAI is about control of temperature, it may be enlightening to see what effect it has (or does not have) on the sinks.

We will add temperature on a second y-axis to 4a.

11. Line 253-254: Cite Trisos et al. 2018 or similar paper about the effects of termination shock when talking about a sudden cessation of SRM having different impacts.
    1. Trisos, C.H., Amatulli, G., Gurevitch, J. *et al.* Potentially dangerous consequences for biodiversity of solar geoengineering implementation and termination. *Nat Ecol Evol* **2**, 475–482 (2018). https://doi.org/10.1038/s41559-017-0431-0

Ok, will do.

12. Line 332: Not necessarily disagreeing with "SAI is not CDR," but this statement is a little unclear. Will the net 60 Tg carbon benefit become net 0 Tg

eventually? If so, then highlight this. If not, then some of SAI's CO2 removal is permanent, if limited. It is worth noting that the paper says in Line 325 that amounts of CO2 of similar magnitude to 60 Gt are "non-negligible."

Our point here is more along the lines of land carbon uptake from SRM not being a permanent and safe storage (in comparison to geological uptakes or MRVed biomass carbon removal). We are going clarify our statement in the text.

13. Line 359: Cirrus Cloud Thinning is technically not Solar Radiation Modification.

While we agree with the reviewer that CCT is strictly seen not SRM, it is usually categorized under Solar Radiation Modification / solar geoengineering approaches, even by the IPCC (Lee et al., 2021).

**Technical Corrections**

1. Line 20: 1.5°C -> 2.0°C

Thank you, we are going to correct it.

2. Line 26 / Line 92: CDR is never defined to be CO2 Removal or Carbon Dioxide Removal; it appears for the first time in line 92.

Thank you for pointing this out. We are going to introduce it in Line 26.

3. Line 33: Injections -> Injection, enhance -> enhances

Ok, we will change it.

4. Line 47: Sur- face

Ok, we will change it.

5. Line 62: "the major levers" feels informal -- consider different phrasing

We are going to change "major levers" to "principal drivers"

6. Figure 1a: mention pre-industrial somewhere (perhaps Y-label or title)

Title changed to "Warming since pre-industrial"

7. Figure 1b: X-label – Years->Year

Ok.

8. Figure 2: mention pre-industrial somewhere (perhaps Y-label or title)

Title changed to "Warming in SSP534-over since pre-industrial"

9. Line 181: The-> the

Since the colon introduces a complete sentence, the first letter is capitalized.

10. Line 195: be- tween

Ok.

11. Line 241: are -> is

Yes, corrected, thank you.

12. Line 251-253: "Hence their call . . . under SRM" is a sentence fragment.

We will add a semicolon before "Hence" and a comma after "Hence"

13. Line 377: "a third *comes* from the ocean."

Thank you, we corrected it.

---

## Author Response (AR1)

**Peer review of "Change in negative emission burden between an overshoot versus peak-shaved Stratospheric Aerosol Injections pathway"**

Journal: Earth System Dynamics
Authors: Susanne Baur, Benjamin M. Sanderson, Roland Séférian, Laurent Terray
Article posted: 5 September 2024
Report submitted: 9 October 2024
General Comments

This study examines stratospheric aerosol injection (SAI) in the CNRM-ESM2-1+ climate model. Specifically, the study considers the SSP5-3.4 "overshoot" scenario with and without an idealized SAI strategy which limits global warming to approximately $2°C$. The authors look at carbon fluxes between the atmosphere and the land and ocean components; because the $CO_2$ concentrations in the atmosphere are fixed throughout, any change in flux to or from the land or ocean must be balanced by a change in anthropogenic emissions. Therefore, changes to these fluxes represent how much additional (or less) anthropogenic emissions are "allowed" in order to follow a given pathway under a given intervention. The authors conclude that, in this experiment, the SAI intervention allows for approximately 60 Gt of additional $CO_2$ emission during SAI ramp-up, 30 Gt during peak deployment, and -30 Gt during phase-out.

The study will be suitable for publication with little additional work. The writing is very high-quality; the text is extremely clear, citations in the introduction and discussion are plentiful, and comparisons to other studies are thorough. I also think the paper identifies an important gap in the literature, as most SAI simulations only run for 30-50 years or until the end of the century, while this study follows SAI through peak deployment, wind-down, termination, and post-deployment out to 2250. However, I would like to see more documentation for the study's methodology, including the process of copying aerosol optical depth (AOD) fields from one model, scaling them up or down, and pasting them into another model, and more acknowledgement that the results may be influenced by the methodology and SAI strategy chosen. I offer specific comments below.

We thank the reviewer for taking the time to go through our work and the constructive feedback. We are pleased to hear that they find the study to be a valuable contribution to the SRM discourse and hope to address their concerns with our modifications.

Specific Comments
Abstract, line 19-20: "SAI is used to maintain $1.5°C$ warming" - I think this is a typo; the rest of the article describes SAI being used to limit warming to $2°C$.

Yes, thank you, we corrected it.

Abstract, lines 20-23 and generally throughout: this study only examines one SAI strategy. While this is perfectly fine (many studies only look at one strategy), the summarized results in the last sentence of the abstract *may* not be applicable to other strategies - e.g., if the initial injection rate were constant instead of ramped up, if the injection happened at different locations, or if the total amount of cooling were different.

Yes, we added to the abstract

> "The findings of this study may be contingent upon the configuration of the injection design and the representation of SAI within the model. Further research is necessary

to validate these results using different models incorporating diverse SAI deployment strategies." (L.26-28)

Line 100, "a global mean temperature increase of 2°C" - relative to what? Temperatures in this study are all presented as an increase relative to some baseline (presumably some preindustrial value), but it is never stated clearly what that baseline is or how it is computed. The baseline value, and its definition, should be stated clearly.

We added "Warming since pre-industrial" to the titles of Figure 1a and Figure 2 and added the definition of temperature baseline in the methods (L.113)

"…baseline on top of which SAI is applied to avoid the temperature overshoot and instead stay at a global mean temperature increase *above pre-industrial (1860-1900)* of 2°C"

Lines 116-117, "The amount of AOD was determined with a trial-and-error approach..." - more needs to be said about how the "injection rates" (AOD magnitude) were chosen. The text offers the explanation that AOD was chosen to limit warming to 2°C ± 0.1°C, but injection starts well before 2°C is reached, and all three ensemble members appear to overshoot the tolerance level around ~2115. The curve in Fig. 1b is very smooth, suggesting that a desired AOD curve was fit, and the shape of the curve tweaked rather than the AOD in each individual year (is 1b a 10-year running mean?). Was AOD chosen first for one ensemble member, and then the same quantity used for the other two ensemble members? This is important because the main message of the paper is "SAI which limits warming to 2°C does X and Y to the carbon cycle," but the magnitude of AOD used doesn't seem to have been chosen purely based on the 2°C target.

We added clarifications on the design of the SAI scenario. In short: Indeed, the AOD curve was fit. 1b is not a 10-year running mean. The same AOD was used for all members, the only difference in the members is the background state of the SSP585 baseline ensemble members. We added:

(L.129-140) "The amount of AOD was determined with a trial-and-error approach guided by the difference in energy balance between the SSP534-over scenario and the global mean radiative characteristics of a SSP126 scenario, which limits warming to 2°C. A curve was fit to the difference in energy balance between the two scenarios and this fitted difference in global mean forcing then translated into spatially resolved AOD using Tilmes et al.'s (2015) G4SSA AOD distribution. The use of the G4SSA AOD distribution is recommended by the GeoMIP protocol for models that cannot dynamically treat sulfur aerosols in the stratosphere (Kravitz et al., 2015) and has been done with the CNRM-ESM model before (Baur et al., 2024a, b; Chen et al., 2023; Jones et al., 2022; Tilmes et al., 2022). See Visioni et al. (2021) for a comparison of the models participating in GeoMIP including CNRM-ESM2-1 with the prescribed G4SSA AOD distribution. G4SSA assumes equatorial injections (Tilmes et al., 2015). However, it has been demonstrated in other models that off-equatorial injection latitudes may perform better at compensating climate change impacts and reducing adverse side-effects from SAI (Kravitz et al., 2019; Tilmes et al., 2017; Visioni et al., 2023).

In this study, AOD was determined for one ensemble member but applied to all three members equally."

Lines 118-119, "The difference in global mean forcing was then translated into spatially resolved AOD using Tilmes et al.'s (2015) G4SSA AOD distribution" - this needs much more explanation. Tilmes, et al. (2015) simulated 8 Tg $SO_2$ injection over the equator during the years 2020-2070 and presented the AOD distribution for that injection strategy. It sounds like this study copied and pasted that AOD distribution into this model, and then scaled it up or down. This is a substantial idealization, and needs to be discussed more - have other models or studies used the AOD fields provided by Tilmes, et al. (2015)? Has it been done in this model before? Has this model done simulations of SAI or volcanic eruptions? The authors should also mention that this AOD distribution corresponds to equatorial SAI, which is relevant because a.) injections at different latitudes could have different results, and b.) equatorial SAI is less commonly studied now because it is known that the aerosols tend to remain confined to the tropical pipe, over-cooling the tropics and under-cooling the poles.

See response to point above. In addition, we added several sentences to the discussion (see last point).

Lines 119-120, "A sufficiently well calibrated SAI magnitude is classified as mostly staying in the range of $2°C \pm 0.1°C$ of warming" - is this your definition, or someone else's?

> (L.141) "A sufficiently well calibrated SAI magnitude is classified as mostly staying in the range of 2°C +/-0.1°C of warming, *as defined by us.*"

Figure 1: it would be very helpful to add atmospheric $CO_2$ concentrations to this figure somehow, either as a separate panel or as a right-hand-side axis on panel (a). They're an integral part of the study's methodology and also play a role in how the three "phases" are defined, but don't appear to be shown anywhere.

Yes, we agree with the reviewer and added CO2 concentration as a second y-axis. To panel a and b.

Lines 144-145, "the results of this study are based on the ensemble mean of the three members except if indicated otherwise" - this should be stated everywhere it is relevant (e.g., in each figure caption) rather than hidden in one place in the text.

We added this information in every figure caption where relevant.

Lines 145-146, "a 10-year rolling mean is displayed on the figures" - is this true for all panels of all figures? If so, this should be stated in each figure caption, rather than buried in the text.

Same as above.

Figure 4b: I recommend adding some kind of errorbar or ensemble spread to this panel.

As an indication of spread we added the single ensemble member differences as crosses around the bars.

Figures 4c-4d: Not strictly necessary, but it would be helpful to the reader to add markers to these panels to denote different points in time - it is not immediately obvious to the reader which end of each curve represents the start of the experiment, what year the "peak" represents, and so on.

Yes, we added descriptions with text and arrows to panel c and d.

Discussion, last paragraph: The authors do a good job of discussing the limitations of using one model and one emissions scenario, but I recommend they also address a.) the idealized nature of the experiment (using prescribed AOD instead of simulating aerosols) and b.) that only one SAI strategy was considered. The same evolution of global mean AOD could have been accomplished with different AOD distributions representing, for example, subtropical or subpolar injection instead of equatorial. Additionally, the 2℃ temperature goal could also have been met with other amounts of AOD; for example, in SSP5-3.4-over, the 2℃ threshold is reached in ~2050 without any SAI - consider an experiment where you used same emissions scenario but with no injection until 2050, at which point SAI is ramped up very suddenly to maintain the 2℃ target. Such a change would probably affect this study's conclusions substantially, while still claiming similar methodology.

Thank you, good point, we added more scenario disclaimers to the discussion and emphasized that the results are influenced by the chosen methodology and SAI strategy. We added:

(L.380-391) "It should be noted that the results of this study are constrained to one Earth System Model. As previous studies have found, carbon cycle processes can vary substantially between different models and increased robustness of the results could be achieved by larger multi-model studies (Plazzotta et al., 2019). Especially models which are able to simulate sulfur injections in the stratosphere rather than using an offline-calculated AOD distribution would be a valuable addition to the literature. Injection design, such as injection timing and varying levels of injection latitude and altitude, has been found to have a significant impact on climate variables at the surface. It can therefore be assumed that different injection strategies would impact the carbon cycle to different degrees. Since the results of this study are constrained to equatorial injections, which have been found to be suboptimal in terms of minimizing adverse side-effects from SAI deployment (Kravitz et al., 2019; Tilmes et al., 2017; Visioni et al., 2023), different injection strategies may lead to different results. Additionally, the 2°C temperature goal could have been met with different injection timing, such as a later start with a more sudden ramp-up of SAI, which could also result in different impacts on the carbon cycle."

**Reviewer #2**

**General Comments**

The paper examines the facilitation and inhibition of CO2 uptake throughout the entire duration of a peak-shaving deployment of SAI. A three-member ensemble simulation of SSP5-3.4-over is compared against SSP5-3.4-sulfur in CNRM-ESM2-1+. The paper finds finds that although there is an enhancement of the carbon cycle during the early phases of the deployment, the later phases see a decline in enhancement, eventually becoming an emission burden in the final phases of peak-shaving and in the years after deployment.

I believe that this paper has valuable contributions and makes some excellent points about the impermanence of SAI's carbon cycle enhancement in a peak-shaving scenario. However, there are portions of the paper in which the language can be imprecise, and the points made are overreaching. In particular, the paper states that phasing out SAI may be made less desirable or more difficult by the burden incurred in the later phases. This statement seems to make certain assumptions about the scenario. In particular, either it assumes that those in the scenario take advantage of the entirety of enhancement provided by SAI in the early stages and ignore that some of SAI's carbon benefit is temporary, or it assumes that those doing CDR in the scenario care more about smaller rate of carbon removal they can do in the later phases than the smaller amount of carbon removal they must do in the later phases.

In all, I believe the science of this paper to be very good, but it draws conclusions from its findings in a way that should be revised.

We thank the reviewer for their valuable comments and recommendation for publication if major points are addressed. We modified the paper to make no statement regarding a scenario's desirability, since it is ultimately a value judgement and, as the reviewer points out, makes assumptions about the scenario. We added more clarity on the temperature baseline, details to plots, an additional paragraph on the areas above and below 0 land and ocean carbon uptake and other little corrections and clarifications the reviewer kindly pointed out.

We hope that our modifications address the reviewers concerns and thank them for a second screening of the paper.

**Specific Comments**

1. Some notes around the concept of making phasing SRM out more difficult:
   1. Line 92: Prolonged SRM deployment and higher CDR in response to sink degradation are a sliding scale, not necessarily an "and". For example, higher CDR to exactly offset the source in phase III of

SSP534-sulfur would result in the same phase out, if I'm understanding correctly.

We added an "or" to emphasize that a sink degradation would prolong SRM deployment **or** require larger amounts of CDR to compensate (L. 105).

2. The statements (e.g. Line 382) that the phase-out of SAI may be made "undesirable" can be further developed, as certain assumptions about those desires can be be made explicit.

   1. If the global community uses up the extra $CO_2$ budget gained from the early phases of peak-shaving by either not mitigating or doing less CDR, then would they have the $CO_2$ concentrations of SSP534-over and thus an extra burden on the tail end. It is relatively clear how the people of Phases II and III might not want to ramp up CDR more than they would have had to.

   2. A less clear example is that if is if they mitigate and do CDR in Phase I as they would have in the baseline SSP534-over case, then they would have lower $CO_2$ concentrations in Phases II and III. They would need more CDR for the same $CO_2$ removal during these phases, but they would also have less $CO_2$ to remove (in net, 60 Gt less). If the speed at which carbon cycle draws out $CO_2$ is the primary desire, they would find difficulty, but if they value getting the $CO_2$ concentration back down to a certain level while maintaining a certain temperature (the premise of peak shaving), I struggle to see how phasing out SAI would be made more undesirable.

The reviewer points out some interesting scenario logics. The conclusions we drew from our results were:

1. Carbon uptake differs under SRM -> what does that mean in terms of negative emission burden
2. Carbon uptake throughout the peak-shaving timeframe varies in a way that it enhances the already pre-existing effect in overshoot scenarios where reducing atmospheric $CO_2$ concentration is "easier" before net-zero $CO_2$ and "harder" afterwards.
   With SRM keeping temperatures at a comfortable level and CDR becoming harder, we concluded that this may make SRM phase-out undesirable. We recognize that this is a subjective statement and completely removed the framing of "desirability" from the paper.

Our conclusions are independent of whether the additional carbon uptake benefit is leveraged or not. Therefore, we refrain from discussing any potential scenario outlooks or assumptions regarding the "use" of the additional carbon budget.

Lines 324-328 / Figure 3: the extra 38 GT of CO2 burden seems like it would be less than two years of extra CDR as modeled in ssp534-sulfur, ending in 2152 instead of 2150. While I agree with the "non-negligible" statement with respect to CDR that exists today, the CDR in the paper's scenarios far outpace the examples the paper gives. This should be at least addressed.

We added:

> (L. 350-351) "Compared to the annual negative emissions assumed in the underlying SSP534 scenario, these additional benefits and burdens appear to be of minor importance. However,…"

2. Line 21: The concept of "buying time" during peak shaving usually refers to time it takes halt temperature rise and reduce it again, and the risks incurred during that temperature peak. It does not usually refer to the extra help from carbon cycle enhancement.

   1. E.g. Zarnetske PL, Gurevitch J, Franklin J, Groffman PM, Harrison CS, Hellmann JJ, Hoffman FM, Kothari S, Robock A, Tilmes S, Visioni D, Wu J, Xia L, Yang CE. Potential ecological impacts of climate intervention by reflecting sunlight to cool Earth. Proc Natl Acad Sci U S A. 2021 Apr 13;118(15):e1921854118. doi: 10.1073/pnas.1921854118. PMID: 33876741; PMCID: PMC8053992.

The "buying time" concept has different interpretations with the most common one being to "reduce pressure" of implementing adaptation and mitigation.

*"Insofar as SAI combats the worst impacts of peak global CO2 emissions, it buys time for different climate approaches that might take longer to show effect, such as mitigation, adaptation or other CDR measures. SAI, as a stop-gap measure with quick results, reduces pressure for implementing other adaptation and decarbonization strategies. It is this reducing pressure framing that seem to lie at the heart of the many versions of buying time or peak-shaving arguments." Neuber & Ott, 2020* ([https://www.mdpi.com/2076-3417/10/13/4637](https://www.mdpi.com/2076-3417/10/13/4637))

3. The difference in compatible emissions do not seem linked to the three phases of SAI deployment specifically, considering the crossover at 2100. Looking at 3b and 4a (for land at least), it looks as though under SSP534-sulfur, the sink becomes more of a sink and the source becomes more of a source. To say in line 377 that the uptake is increased during SAI roll-out but decreased during phase-out feels off, since it seems to have more to do with

when it's a source or sink, and thus the background GHG emissions pathway and the CO2 concentrations.

1. In other words, would phasing out SAI while doing no CDR cause the negative difference starting at 2100 in Figure 3b? Alternatively, would not phasing out SAI and maintaining a constant amount of cooling while doing the same amount of CDR as simulated create the 2100 crossover?

We agree with the reviewer that the response of the land reservoir does not correlate directly with the stages of SAI deployment. We made sure that throughout the paper it is clear that the C uptake is enhanced when land acts as a sink and reduced when land acts as a source. But that the primary mechanism of whether it is a sink or driver depends on the CO2 concentration, not SAI.

(L.370-380) "This transition from enhanced uptake to additional outgassing around the year 2100 corresponds temporally with the shift from land carbon sink to land carbon source under SSP534-sulfur and -over (Fig. 4a). This observation suggests that SAI does not cause the transition from sink to source, but rather reinforces the signal, enhancing uptake compared to the no-SAI scenario when it acts as a sink, and enhancing outgassing when it acts as a source. As the transition from source to sink occurs at similar points in time in both scenarios, the underlying atmospheric CO2 concentration clearly plays a dominant role in the sink dynamics. However, the timing of the peak phase down of SAI is coincident with the shift from natural sink to source in SSP534-over, so the dynamical evolution of SAI in this experiment potentially causes two effects which cannot easily be disambiguated using this single experiment: SAI could potentially be enhancing the CO2-driven sink dynamics, but also the phase-out of SAI may be itself causing a shift from carbon sink to source. Disambiguating these factors requires additional scenarios, such as a configuration that removes the negative emission phase while maintaining the same SAI deployment."

(Abstract L.20-25): "The results indicate that carbon effects are reinforced under SAI. While the land carbon reservoir is a carbon sink, SAI enhances the uptake further; when the land acts as a carbon source, SAI enhances the outgassing. Thereby, carbon fluxes associated with SAI evolve over time: The increase in carbon uptake under SAI during the positive emission phase confirms prior studies and substantiates the concept of buying time during SAI ramp-up, later stages of the peak-shaved SAI scenario show the carbon benefit reducing and turning into an additional obstacle making a phase-out of SAI more difficult by enhancing the carbon removal burden"

However, we do not find anything wrong with our sentence in L.425: The land sink is more dynamic to changes in SAI, as uptake is substantially increased during SAI rollout, but reduced during parts of SAI phase-out and post-deployment, whereas ocean sink is slightly enhanced during the roll-out period but stays close to the overshoot baseline thereafter.

4.  Line 26: Although CO2 removal is often considered a form of mitigation (IPCC AR6 WGIII: CDR Factsheet), it is also often considered separate from mitigation (e.g. NASEM, 2021: "This portfolio must involve reducing GHG emissions to the atmosphere (mitigation), and removing carbon from the atmosphere and reliably sequestering it"). This paper should explicitly state whether it is defining mitigation to include CDR or considering CDR to be independent, cite a source to support that definition, and then stick to it for the duration of the paper.

    1.  I would recommend separating the two, such that something like the first line of the abstract "…allowing additional time for the implementation of conventional climate mitigation strategies and CO2 removal." could be read without confusion, regardless of whether the reader defines CO2 removal as mitigation or not.

We thank the reviewer for pointing this out. We removed the term "conventional mitigation" and instead clarified that we refer to emission reductions and CDR with the term mitigation, aligned with the denomination of the latest IPCC report, that classifies CDR as a form of mitigation (Riahi et al., 2022).

(L.30) "Solar Radiation Modification (SRM) is increasingly being discussed as a potential temporary approach to lower global mean temperature while mitigation efforts, such as emission reductions and atmospheric Carbon Dioxide Removal (CDR), are being sufficiently scaled up (Climate Overshoot Commission, 2023; NASEM, 2021)."

5.  Lines 37-78: Be clear about the baseline that the paper is comparing SAI to with each comparison. For example, the CO2 fertilization effect is likely to the same temperature, no-SAI, but the reduced heat stress is likely to the same CO2 concentration or year, no-SAI.

We agree that this is important, as highlighted in L. 317-324, and included information on the comparison baselines in L.42-91.

6.  Line 100: Regarding "increase of 2°C", say relative to what baseline (pre-industrial, but also discuss how are it is being defined)

We added "Warming since pre-industrial" to the titles of Figure 1a and Figure 2 and added the definition of temperature baseline in the methods (L.113)

"...baseline on top of which SAI is applied to avoid the temperature overshoot and instead stay at a global mean temperature increase *above pre-industrial (1860-1900)* of 2°C"

7. Line 117: The paper should give SSP126 the same background as it gave SSP534. At present, it is put into the text without definition.

We disagree with the reviewer's judgment here since our scenarios have nothing to do with the SSP126 storyline itself and no comparisons of our simulations with a SSP126 pathway are made. We are going to add the words "the global mean radiative characteristics of a SSP126 pathway" in L 130 to add clarification.

8. Figure 3b: I am curious about the total area over 0 and under 0 (which sum to be 60 Gt), especially since there seem to be two "phases": where the difference is positive going until ~2100 and where it is negative after.

We added:

(L.368) "It is evident from Figure 3b that there are two distinct periods for carbon uptake differences under SAI. The first period extends until 2100, during which an additional 143 Gt $CO_2$ are taken up by terrestrial and marine reservoirs under SAI. The subsequent period goes until around 2220 where 83 Gt $CO_2$ less are taken up under SAI than the overshoot scenario. This transition from enhanced uptake to additional outgassing around the year 2100 corresponds temporally with the shift from land carbon sink to land carbon source under SSP534-sulfur and -over (Fig. 4a)."

9. Figure 3b: It would be nice to have a measure of variability – what is significant difference between compatible emissions vs. what is a byproduct of variability.

We added the single members results as fine lines around the mean.

10. Figure 4: Some sort of Carbon sink vs. Temperature plot may be useful here. Masking temperature by land and ocean could be included, or use cooling done by SAI vs. difference in sinks, but given that peak-shaving SAI is about control of temperature, it may be enlightening to see what effect it has (or does not have) on the sinks.

Good point. We added temperature difference between the scenarios / SAI induced cooling as a second y-axis to 4a.

11. Line 253-254: Cite Trisos et al. 2018 or similar paper about the effects of termination shock when talking about a sudden cessation of SRM having different impacts.

1. Trisos, C.H., Amatulli, G., Gurevitch, J. *et al.* Potentially dangerous consequences for biodiversity of solar geoengineering implementation and termination. *Nat Ecol Evol* **2**, 475–482 (2018). https://doi.org/10.1038/s41559-017-0431-0

Done. L.279

12. Line 332: Not necessarily disagreeing with "SAI is not CDR," but this statement is a little unclear. Will the net 60 Tg carbon benefit become net 0 Tg eventually? If so, then highlight this. If not, then some of SAI's CO2 removal is permanent, if limited. It is worth noting that the paper says in Line 325 that amounts of CO2 of similar magnitude to 60 Gt are "non-negligible."

Whether the 60Tg carbon benefit will become 0Tg or not cannot be said from our study. Our point in these sentences refers to the fact that any additional land carbon uptake from SRM is volatile and cannot compare to the permanence and storage safety of monitored CDR.

L.357 "Given the variability of the terrestrial carbon fluxes in this study and the storage safety and storage timescales considered in common CDR technologies, this study supports the statement that the carbon cycle enhancement during peak-shaving SAI is volatile and transient and cannot be referred to as CDR and counted as such."

Line 359: Cirrus Cloud Thinning is technically not Solar Radiation Modification.

While we agree with the reviewer that CCT is strictly seen not SRM, it is usually categorized under Solar Radiation Modification / solar geoengineering approaches, see the latest IPCC (Lee et al., 2021).

**Technical Corrections**

1. Line 20: 1.5°C -> 2.0°C

Thank you, we corrected it.

2. Line 26 / Line 92: CDR is never defined to be CO2 Removal or Carbon Dioxide Removal; it appears for the first time in line 92.

Thank you for pointing this out. L.32

3. Line 33: Injections -> Injection, enhance -> enhances

Done.

4. Line 47: Sur- face

Done.

5. Line 62: "the major levers" feels informal -- consider different phrasing

We changed "major levers" to "principal drivers"

6. Figure 1a: mention pre-industrial somewhere (perhaps Y-label or title)

Title changed to "Warming since pre-industrial"

7. Figure 1b: X-label – Years->Year

Done.

8. Figure 2: mention pre-industrial somewhere (perhaps Y-label or title)

Title changed to "Warming in SSP534-over since pre-industrial"

9. Line 181: The-> the

Since the colon introduces a complete sentence, the first letter is capitalized.

10. Line 195: be- tween

Done.

11. Line 241: are -> is

Yes, corrected, thank you.

12. Line 251-253: "Hence their call . . . under SRM" is a sentence fragment.

We added a semicolon before "Hence" and a comma after "Hence"

13. Line 377: "a third *comes* from the ocean."

Thank you, we corrected it.